# Rapid evolutionary change in trait correlations of single proteins

Pouria Dasmeh [1,2,3,10] ✉, Jia Zheng [4,5,6,10], Ayşe Nisan Erdoğan[7], Nobuhiko Tokuriki [7] & Andreas Wagner [2,3,8,9] ✉

Many organismal traits are genetically determined and covary in evolving populations. The resulting trait correlations can either help or hinder evolvability – the ability to bring forth new and adaptive phenotypes. The evolution of evolvability requires that trait correlations themselves must be able to evolve, but we know little about this ability. To learn more about it, we here study two evolvable systems, a yellow fluorescent protein and the antibiotic resistance protein VIM-2 metallo beta-lactamase. We consider two traits in the fluorescent protein, namely the ability to emit yellow and green light, and three traits in our enzyme, namely the resistance against ampicillin, cefotaxime, and meropenem. We show that correlations between these traits can evolve rapidly through both mutation and selection on short evolutionary time scales. In addition, we show that these correlations are driven by a protein's ability to fold, because single mutations that alter foldability can dramatically change trait correlations. Since foldability is important for most proteins and their traits, mutations affecting protein folding may alter trait correlations mediated by many other proteins. Thus, mutations that affect protein foldability may also help shape the correlations of complex traits that are affected by hundreds of proteins.

Evolvability is a biological system's ability to bring forth novel and adaptive phenotypes. Because evolvability varies among organisms and traits, it can itself evolve[1]. Understanding the factors that affect its evolution matters not only for our fundamental understanding of biological evolution. It also matters for technological applications, including the experimental evolution of novel and useful molecules, such as efficient industrial enzymes[2].

Many phenotypic traits are correlated with one another, and these correlations often have a genetic basis[3–5]. Examples of such traits include the weight and the height of individuals within a population[6]. They also include coloration and behavioral traits in many animal species[3,7,8], as well as seed dormancy and flowering time in plants[9]. Genetic correlations between phenotypic traits can influence evolvability, because they can affect the extent to which traits can evolve independently from one another[10–14].

On the one hand, correlated trait evolution can facilitate evolvability. For example, a strong correlation between mouth and jaw morphology in species of freshwater fish from the genus *Micropterus* helps these species adapt to the consumption of large prey[15,16]. On the other hand, strong trait correlations can also hamper evolvability and render weak trait correlations advantageous. For example, a weak correlation between early tetrapod forelimb and hindlimb

[1]Center for Human Genetics, Marburg University, Marburg 35043, Germany. [2]Institute for Evolutionary Biology and Environmental Studies, University of Zurich, Zurich 8057, Switzerland. [3]Swiss Institute of Bioinformatics (SIB), Lausanne 1015, Switzerland. [4]Zhejiang Key Laboratory of Structural Biology, School of Life Sciences, Westlake University, Hangzhou 310030, China. [5]Westlake Laboratory of Life Sciences and Biomedicine, 310030 Hangzhou, China. [6]Institute of Biology, Westlake Institute for Advanced Study, 310030 Hangzhou, China. [7]Michael Smith Laboratories, University of British Columbia, Vancouver, BC V6T 1Z4, Canada. [8]The Santa Fe Institute, Santa Fe, New Mexico 87501, US. [9]Stellenbosch Institute for Advanced Study (STIAS), Wallenberg Research Centre at Stellenbosch University, Stellenbosch 7600, South Africa. [10]These authors contributed equally: Pouria Dasmeh, Jia Zheng. ✉ e-mail: dasmeh@staff.uni-marburg.de; andreas.wagner@ieu.uzh.ch

morphology helped bring forth major specializations, such as bipedalism and flight[17,18]. More generally, the decoupling of traits from one another helps trait variation become individuated within a population, such that genetic change can affect only one trait without affecting others. Trait individuation can help explain phenomena as different as the evolution of different cell types in multicellular organisms[19], leaf shapes in plants[20], and body structures in animals[21].

Because trait correlations are important for evolvability, it is important to study how easily these correlations can themselves evolve. Previous studies have shown that trait correlations can change on long evolutionary time scales, and become weaker or stronger in some lineages of different species. For example, the correlation between beak and skull morphology in Darwin's Galapagos finches (genus *Geospiza*) and Hawaiian honeycreepers (*Drepanis coccinea*) is significantly stronger compared to their continental relatives[22], which has facilitated rapid craniofacial evolution in the former species. Another example comes from indigenous Australian dicotyledons – plants with two embryonic leaves – where the correlations between several seedling traits such as the appearance of scale-like leaves are evolutionarily malleable across thousands of species[23]. Although examples like these show that trait correlations can evolve, we know little about how rapidly they can do so. In addition, we cannot easily quantify the contribution of mutations and selection to these correlations and their evolutionary change. The reason is that these traits are complex, macroscopic features of multicellular organisms, which may be influenced by dozens or hundreds of genes. Also, it is difficult to determine the causes of correlations between complex traits. In principle, they can be caused by genetic factors, such as pleiotropy and linkage disequilibrium[24], or by environmental and ecological factors, such as habitat temperature and aridity[25,26].

To overcome some of these challenges, we study evolutionary change in trait correlations in single protein-coding genes. For this purpose, we chose two genes that we had previously subjected to experimental evolution, and whose traits we can measure reliably. The first is the gene *yfp*, which encodes yellow fluorescent protein (YFP). Because this protein is not native to the microbial host organism *E. coli* in which we study it, we can study its traits with less interference from the host's proteome and physiology than would be possible for native proteins. YFP emits both yellow and green fluorescence light, and these emissions are the two traits we study (Fig. 1A). Although these traits are not directly linked to cellular fitness, they can change by mutation and selection. We can quantify them rapidly and precisely in thousands of organisms through fluorescence activated cell sorting (FACS), which allows us to precisely estimate trait correlations.

The second gene extends our analysis to traits that are directly linked to cellular fitness. It is the gene *vim2*, which encodes the protein

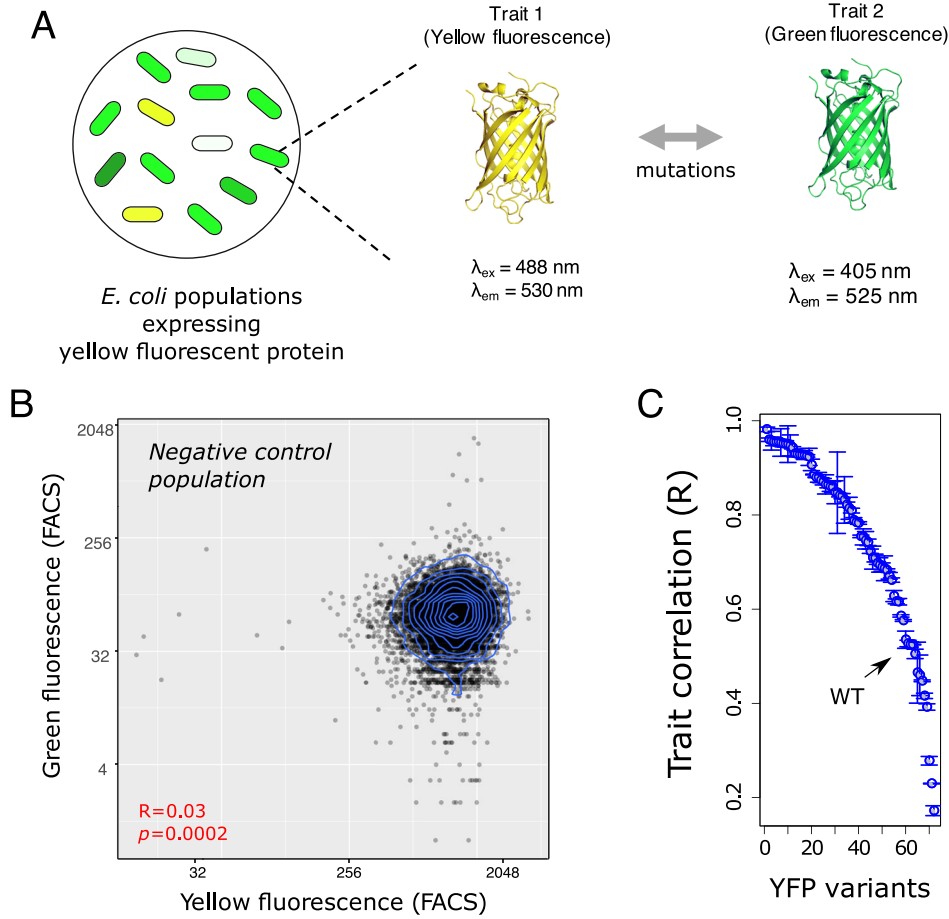

**Fig. 1 | Gene expression noise and mutations can change the strength of correlation between yellow and green fluorescence intensities. A** Schematic of experimental design. We investigate the correlation between yellow and green fluorescence intensities in *E. coli* populations that express yellow fluorescent protein (YFP). **B** Autofluorescence correlation between yellow and green fluorescence intensities in an *E.coli* population without the *yfp* gene. Blue lines show the contours of a two-dimensional density of data points. **C** The correlation between yellow and green fluorescence intensities varies substantially among YFP mutants. Each circle corresponds to the mean trait correlation for a YFP mutant, and error bars show one standard deviation of Spearman's rank correlation coefficient between yellow and green fluorescence intensities in three biological replicates of populations expressing each mutant. Data in panels **B** and **C** are based on fluorescence-activated cell sorting (FACS) with ~$10^5$ sorted cells (see Methods for details). Source data for panels **B** and **C** are available in the Source Data file.

VIM-2 metallo betalactamase (MβL). VIM-2 metallo-β-lactamase is a highly effective enzyme that confers broad-spectrum resistance against beta-lactam antibiotics. VIM-2 belongs to the genetically and functionally diverse MβL superfamily, which has the remarkable ability to efficiently hydrolyze distinct classes of β-lactam antibiotics[27–29]. Given its broad-spectrum resistance capabilities, VIM-2 is an ideal candidate for experiments on trait correlations. The VIM-2 traits that we study are resistance to the three different antibiotics ampicillin, cefotaxime, and meropenem. We have previously shown that only a few mutations suffice to change the resistance conferred by VIM-2 to these antibiotics[27,30]. Here we quantify correlations between these antibiotic resistance traits.

Using both *yfp* and *vim2*, we investigate whether trait correlations can rapidly change by mutations and selection at the level of single genes. We demonstrate that trait correlations are malleable and can undergo substantial changes on short evolutionary time scales. Although the extent of these changes varies between the proteins we study, they are predominantly affected by changes in the biophysical properties of these proteins.

## Results

### Trait correlation changes by mutations in fluorescent proteins

We first examined trait correlations in our fluorescent protein, YFP, and then extended our observations and findings to the enzyme VIM-2. Single point mutations can shift the maximum emission wavelength of YFP. This property facilitates the experimental evolution of fluorescence color in YFP-expressing cell populations (Fig. 1A). In addition, it allowed us to ask whether mutation and/or selection can change the correlation between yellow and green fluorescence. To do so, we systematically measured this correlation in both genetically polymorphic populations of *E.coli* cells that express different YFP variants, as well as in several YFP mutants. In addition, we also measured this correlation in YFP subpopulations that differ in their fluorescence intensities. These experiments enabled us to distinguish the extent to which microenvironmental variation (gene expression noise), mutation, and selection affect the correlation between two traits of a single protein.

Before starting our main experiments, we needed to make sure that the trait correlation we measure and study in this work is only the property of fluorescent proteins and not the background-fluorescence resulting from other fluorescing molecules[31]. We thus measured both the yellow and green autofluorescence of *E.coli* cells that do not express YFP, and found that the two autofluorescence traits are only weakly correlated (Spearman's rank correlation $R = 0.03$, $p = 0.0002$; Fig. 1B). We consider this correlation our baseline correlation. Any significantly stronger correlations in YFP-expressing cells can be attributed to YFP, and not to cellular autofluorescence.

We next turned to our first focal question: Can mutations strengthen or weaken this baseline correlation. To answer this question, we measured trait correlations for 71 YFP mutants that we had previously engineered, because they attained moderate to high frequency in evolving populations[32,33]. These variants include the WT protein, as well as 10 mutants with one, 28 mutants with two, and 32 mutants with three amino acid changes (Supplementary Table 1). All double mutations share the mutation G66S (replacement of a glycine with serine at position 66 of YFP) or Y204C, and all triple mutations share both amino acid changes G66S and Y204C. The mutations G66S and Y204C are unique in that they shift the emission spectrum of YFP from yellow towards green fluorescence[32], reducing yellow fluorescence but enhancing green fluorescence. The nature of these mutations shows that yellow and green fluorescence are two distinct features of YFP and can be treated as separate (albeit possibly correlated) traits.

Trait correlations varied significantly among YFP mutants (Fig. 1C). Specifically, they varied from $R = 0.17$ (for the triple mutant G66S-Y204C-F72S; $p < 10^{-16}$, Spearman's rank correlation) to $R = 0.98$

(for the double mutant G66S-N145S, $p < 10^{-16}$; Spearman's rank correlation). Remarkably, even some single-point mutations sufficed to substantially strengthen or weaken the correlations of these traits relative to the WT protein. We further investigated whether changes in trait correlations arise from changes in the proportion of cells exhibiting autofluorescence or low fluorescence intensity. We found instead that they stem from variation in the fluorescence intensity of functional and actively fluorescing molecules (Supplementary note 1). Altogether, these observations show that trait correlations in our system can change substantially by single-point mutations.

### Selection of one trait changes trait correlation in fluorescent proteins

We next addressed our second focal question: Can selection change trait correlations? To answer it, we measured the correlation between yellow and green fluorescence intensities in YFP populations that we had previously evolved under multiple cycles of mutation and either strong selection, weak selection, or no selection for yellow fluorescence intensity[32,33] (Fig. 2A, see Methods for details). Selection leads to the accumulation of different YFP mutants in these population. The properties of such variants differ from that of wild-type (WT) YFP. To quantify these differences, we calculated the ratio of the median yellow fluorescence of different populations to that of an isogenic population of YFP wild-type. For populations under no selection this ratio was approximately 0.68. For populations under weak selection it was approximately 1, and for populations under strong selection it was 18. Previous single-molecule real-time sequencing had also shown that during experimental evolution, YFP accumulated up to ~6 amino acid changes compared to the wild-type[32].

During experimental evolution, the trait correlation increased from $R = 0.13$ ($p < 10^{-16}$, Spearman's rank correlation) for YFP populations that had evolved under no selection, to $R = 0.73$ and $R = 0.89$ ($p < 10^{-16}$, Spearman's rank correlation), for populations that had evolved under weak and strong selection, respectively (Supplementary Fig. 1). Because we suspected that trait correlations may depend on absolute fluorescence intensities, we pooled YFPs from these three populations and further partitioned this pooled population into 20 subpopulations, such that each subpopulation spanned a similar interval of yellow fluorescence intensity (Fig. 2B; see Methods). We then sorted ~$10^5$ cells in each subpopulation using fluorescence-activated cell sorting (Fig. 2B), and measured the correlation between yellow and green fluorescence intensities in each subpopulation.

Remarkably, the correlation between yellow and green fluorescence varied dramatically among the 20 subpopulations. It increased from a weak correlation ($R = 0.03$, $p = 0.00023$; Spearman's rank correlation) for the first subpopulation with the lowest average yellow fluorescence intensity, to a strong correlation ($R = 0.87$, $p < 10^{-16}$; Spearman's rank correlation) for the 20th subpopulation with the highest average yellow fluorescence intensity (Fig. 2C–E). The correlation between yellow and green fluorescence was significantly different ($p < 0.05$; Fisher's z-transformation, Methods) from that caused by mutations (Spearman's $R = 0.63$) for all except the 12th subpopulation ($p - 0.3$, Fisher Z-transformation). These results show that selection can systematically change the strength of a trait correlation, and render it significantly stronger or weaker than the one caused by random mutations. To further validate this observation, we picked 90 single clones from each of the 20 subpopulations and measured their yellow ($\lambda_{ex} = 485$ nm, $\lambda_{em} = 530$ nm) and green ($\lambda_{ex} = 400$ nm, $\lambda_{em} = 512$ nm) fluorescence intensities using a microplate reader (TECAN Spark) (Fig. 2F, see Methods for details). For these single clones too, we observed an increase in the correlation of the green and yellow fluorescence intensities from the first subpopulation to the 20th subpopulation (Fig. 2G, H). Altogether, our observations show that trait correlations can be easily shaped by selection, even on the short time scales of laboratory evolution.

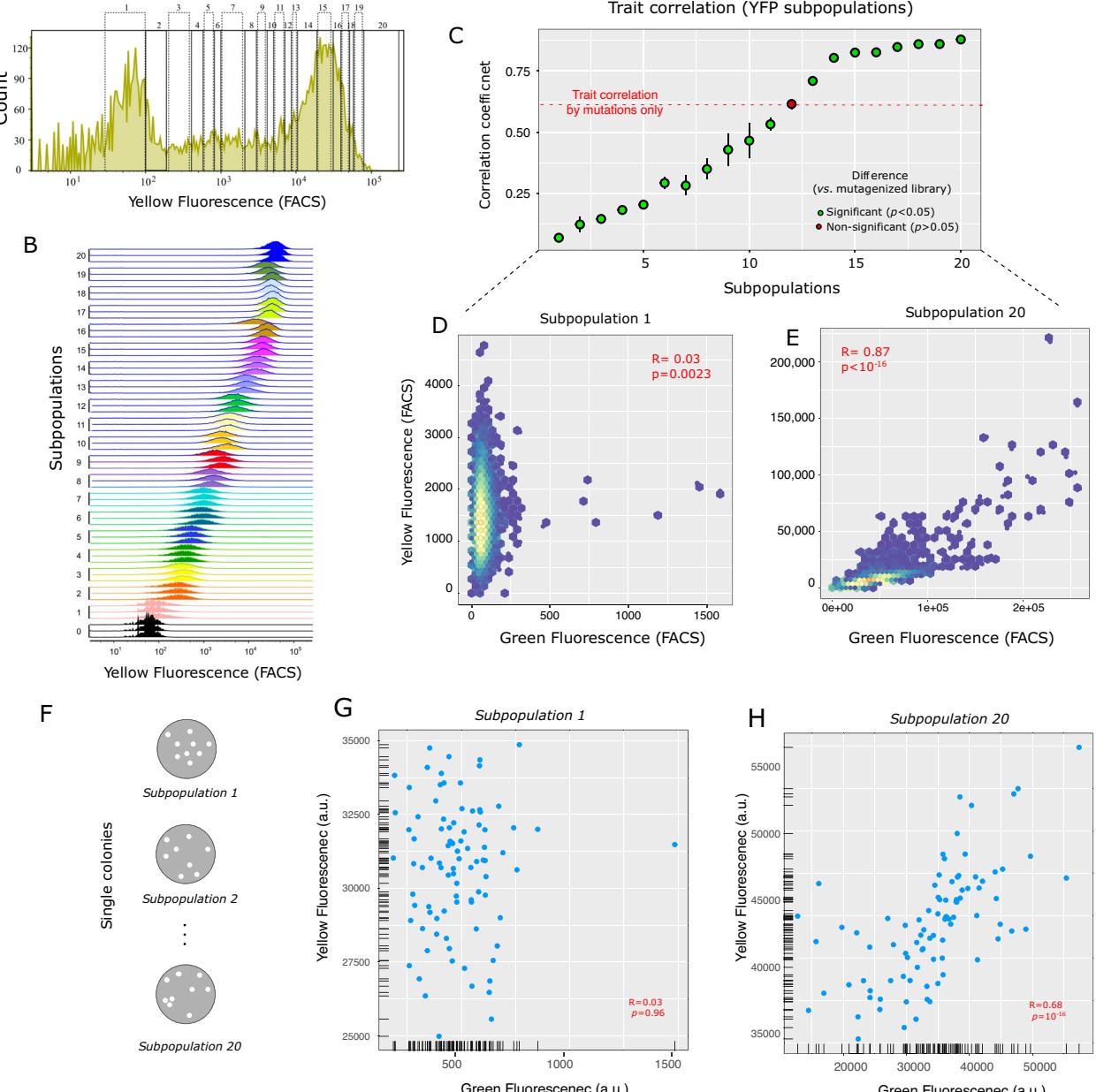

**Fig. 2 | Selection systematically changes the trait correlation measured at the level of single cells. A** Distribution of fluorescence intensity from fluorescence-activated cell sorting (FACS) of our pooled population, which contains YFP variants that have evolved under mutation and various strengths of selection to maintain yellow fluorescence (see "Methods"), and that display different fluorescence intensities. The vertical axis indicates the number of cells at a given yellow fluorescence intensity (horizontal axis, arbitrary units). To study cells of different fluorescence intensity, we sorted cells from this population into twenty bins (subpopulations) according to their yellow fluorescence intensity (Subpopulations 1–20, as indicated by lettering on top of each bin). **B** Each of the subpanels from bottom to top shows the distribution of fluorescence intensities of cells in one of the 20 subpopulations (right vertical axis). **C** The mean Spearman's correlation coefficient for the correlation between yellow and green fluorescence intensities in all 20 subpopulations (indicated on the x-axis). Error bars represent the standard deviation of the correlation coefficient, as measured from three replicate samples of the same subpopulation. Green circles correspond to subpopulations in which the correlation between yellow and green fluorescence was significantly different from that caused by single-point mutations ($p < 0.05$). The red circle and bar

correspond to the 12th subpopulation, in which trait correlation was statistically indistinguishable from that caused by random mutations ($p < 0.05$). We compared the significance of the trait correlation in each subpopulation with the correlation caused by mutations ($R = 0.63$, $p < 10^{-16}$, Spearman's rank correlation), using Fisher's z-transformation (see Methods). The p-values of this comparison for all subpopulations are -0, -0. -0, -0, $2.69 \times 10^{-290}$, $9.46 \times 10^{-191}$, $4.63 \times 10^{-185}$, $1.11 \times 10^{-122}$, $6.80 \times 10^{-65}$, $5.37 \times 10^{-30}$, $5.44 \times 10^{-17}$, 0.30, $1.04 \times 10^{-20}$, $5.82 \times 10^{-132}$, $6.90 \times 10^{-186}$, $1.33 \times 10^{-194}$, $7.71 \times 10^{-261}$, -0, $1.11 \times 10^{-304}$, and -0. **D** Yellow fluorescence intensity versus green fluorescence intensity for the first subpopulation (with lowest fluorescence intensity). **E** Yellow fluorescence intensity versus green fluorescence intensity for the 20th subpopulation (with the highest fluorescence intensity). **F** We picked 90 single clones from each of our twenty subpopulations, and measured the correlation between yellow and green fluorescence intensities in liquid cultures derived from each of these 1800 (=90 × 20) colonies (see "Methods"). Yellow fluorescence intensity versus green fluorescence intensity for the 90 clones sampled from **G** the first subpopulation and **H** the 20th subpopulation. All correlation coefficients (R) are Spearman's rank correlations. Source data for panels **C**–**E**, **G**, and **H** are available in the Source Data file.

## Protein foldability contributes to the malleability of trait correlation

We next asked which YFP properties can (i) be easily shaped by one or few mutations, and (ii) affect the trait correlations we observe in individual variants and in our polymorphic subpopulations. One candidate property is fluorescence color itself. Mutations in YFP, particularly those that alter amino acids close to the protein's fluorophore, can shift the emission spectrum of the protein. We compared trait correlations for a set of 10 YFP variants that maximally emit in colors ranging between yellow (~525–530 nm) and green (~510 nm; see Supplementary Table 2). However, a change in this maximum emission wavelength was not significantly associated with an altered trait correlation ($R = 0.28$, $p = 0.42$; Spearman's rank correlation). For example, although the maximum emission wavelength was substantially different between the mutant G66S (~521 nm) and the triple mutants Y204C-G66S-F65L (~511 nm), the trait correlation is nearly the same in these two mutants (~0.9). Another example involves the triple point mutants that contain both color-shifting mutations Y204C and G66S (~511 nm). Among all such mutants, trait correlation varied from 0.68 (for Y204C-G66S-K102E) to 0.87 (Y204C-G66S-F65L). These observations show that the fluorescence color itself. i.e., the emission spectrum, is not a key property affecting the trait correlations we study.

We then focused on a second candidate property, which is biophysical in nature. It is the ability of a protein to fold properly. We hypothesized that changes in protein foldability can affect trait correlations, because only folded proteins fluoresce. To validate this hypothesis, we estimated the foldability of our 10 YFP variants. Specifically, we measured the overall refolding yield of YFP upon thermal denaturation (see Methods). In this assay, we first denatured YFP, allowed it to refold, and quantified the amount of refolded proteins by measuring the fluorescence relative to the fluorescence of YFPs that had not denatured[32]. Importantly, more foldable YFP variants showed a significantly higher correlation between green and yellow fluorescence intensities (Fig. 3B; $R = 0.67$, $p = 0.03$; Spearman's rank correlation). Some of our 10 YFP variants in this analysis harbored the known foldability-improving mutations F47L, V164A, and F65L[32]. The presence of these mutations alone increased the trait correlation. For example, the trait correlation of the variant G66S-Y20C increased from ~0.61 to ~0.75, 0.78, and 0.87, in the presence of the mutation F47L, V164A and F65L, respectively (Supplementary Table 2).

To further validate the hypothesis that changes in foldability determine evolutionary changes in trait correlations, we turned to our 20 subpopulations and quantified the overall refolding yield of YFP upon thermal denaturation for these subpopulations. We also measured the temperature of midpoint denaturation ($T_m$) as a measure of thermodynamic stability, another quantity that correlates well with protein foldability[34]. Both measures of protein foldability systematically increased from the first to the 20th subpopulation (Fig. 3C, D and Supplementary Fig. 2). Both measures were also themselves highly associated with trait correlations (Fig. 3E–G; Spearman's $R = 0.93$ and 0.99 for $T_m$ and refolding yield after thermal denaturation, respectively; $p \sim 10^{-6}$). In addition, we also used an enzyme-linked immunosorbent assay (ELISA) to measure the amount of soluble protein in each subpopulation, which is also a proxy for protein foldability. Indeed, more soluble YFP subpopulations also fluoresced more intensely (Supplementary Fig. 2). More importantly, protein solubility was again strongly associated with the magnitude of our trait correlations ($R = 0.99$, $p \sim 10^{-6}$; Spearman's rank correlation). Altogether, these results show that protein foldability is a key determinant of the correlation between the two traits we study in YFP.

## Trait correlations changes in the evolution of an enzyme

So far, we have shown that correlations in color emission of fluorescent proteins change by mutations and selection and are shaped by protein foldability. Would these conclusions also apply to a different protein whose traits are directly linked to cellular fitness? To find out, we studied the VIM-2 metallo-β-lactamase (MβL), a highly effective enzyme that confers broad-spectrum resistance against beta-lactam antibiotics. (Fig. 4A)[27,29]. Given its broad-spectrum resistance conferring abilities, VIM-2 is an ideal candidate for our analysis of trait correlation. Here, we consider resistance to the three different beta-lactam antibiotics ampicillin, cefotaxime, and meropenem as our traits of interest.

Just as in the case of fluorescent proteins, we first investigated whether mutations could change trait correlations. Assessing trait correlation in YFP and VIM-2 populations requires different measurement methods: unlike light emission, antibiotic resistance cannot be measured in single cells. We thus compared trait correlations across different genotypes from deep mutational scanning experiments. This is akin to averaging the individual differences in single cells of a given genotype, and doing so for multiple genotypes. For this analysis, we analyzed data from deep mutational scans that we had previously performed to assess the fitness effect of single-point mutations in VIM2 on E. coli's resistance to our three antibiotics[27,29].

As shown in Fig. 4B, C, changes in antibiotic resistance (our fitness measure, see Methods) of VIM-2 mutants relative to WT VIM-2 are correlated across the pairs of antibiotics. On average, a mutation that increases resistance against ampicillin also tends to enhance resistance to cefotaxime and meropenem. The strength of this association was ~0.83 and ~0.94 (Spearman's rank correlation) between ampicillin and cefotaxime, and between ampicillin and meropenem, respectively. Importantly, the trait correlation was weaker for more deleterious (resistance-reducing) than for less deleterious mutations. To demonstrate this, we divided all mutations into two bins based on how they affect ampicillin resistance. The first bin encompassed mutations that are highly deleterious (fitness scores −12 to −4, referred as 'bin 1' in Fig. 4B, C) and cause the ampicillin MIC of the enzyme to fall below the intrinsic resistance of E. coli (4 μg/mL). The second bin encompassed mutations above this resistance threshold (fitness scores −4 to 2, referred as 'bin 2'). For ampicillin and cefotaxime resistance, the trait correlation in the first bin was ~0.70 (Spearman's correlation, $p < 0.0001$). It increased to ~0.91 (Spearman's correlation, $p < 0.0001$) in the second bin. Likewise, for ampicillin and meropenem resistance, the trait correlation within the first bin was ~0.71 (Spearman's correlation, $p < 0.0001$), and increased to ~0.91 (Spearman's correlation, $p < 0.0001$) for the second bin. These changes in correlation strength were significant, as indicated from a Fisher transformation of Spearman's rank correlation ($p < 0.001$), showing that mutations can significantly change trait correlations in VIM2.

Next, we investigated whether significant changes in trait correlations occur within VIM-2 populations that are subject not just to mutations but also to selection. To this end, we took advantage of a laboratory evolution experiment that we had performed recently. In this experiment we evolved VIM-2 through 100 rounds of mutagenesis and selection under a fixed and low ampicillin concentration (10 μg/ml), starting from a VIM-2 variant with high resistance against ampicillin (MIC = 8192 μg/ml, compared to MIC = 128 μg/ml in WT VIM-2; Fig. 1D, Supplementary Fig. 4, and S5)[35]. The concentration of ampicillin and thus the strength of selection remained constant in this experiment. Since the experiment started from very high resistance, deleterious mutations gradually accumulated and reduced the average ampicillin resistance of the evolving population, until the population reached a mutation-selection equilibrium. In this equilibrium population members differed widely in their mutation load and their antibiotic resistance. Specifically, the VIM-2 variants at the end of this experiment genotypically differed by ~15–75 amino acid mutations from wild-type VIM-2, and phenotypically differed by up to 13-fold in their ampicillin minimum inhibitory concentration (MIC). We examined the correlation between resistance against the three beta-lactam antibiotics at different time points during evolution.

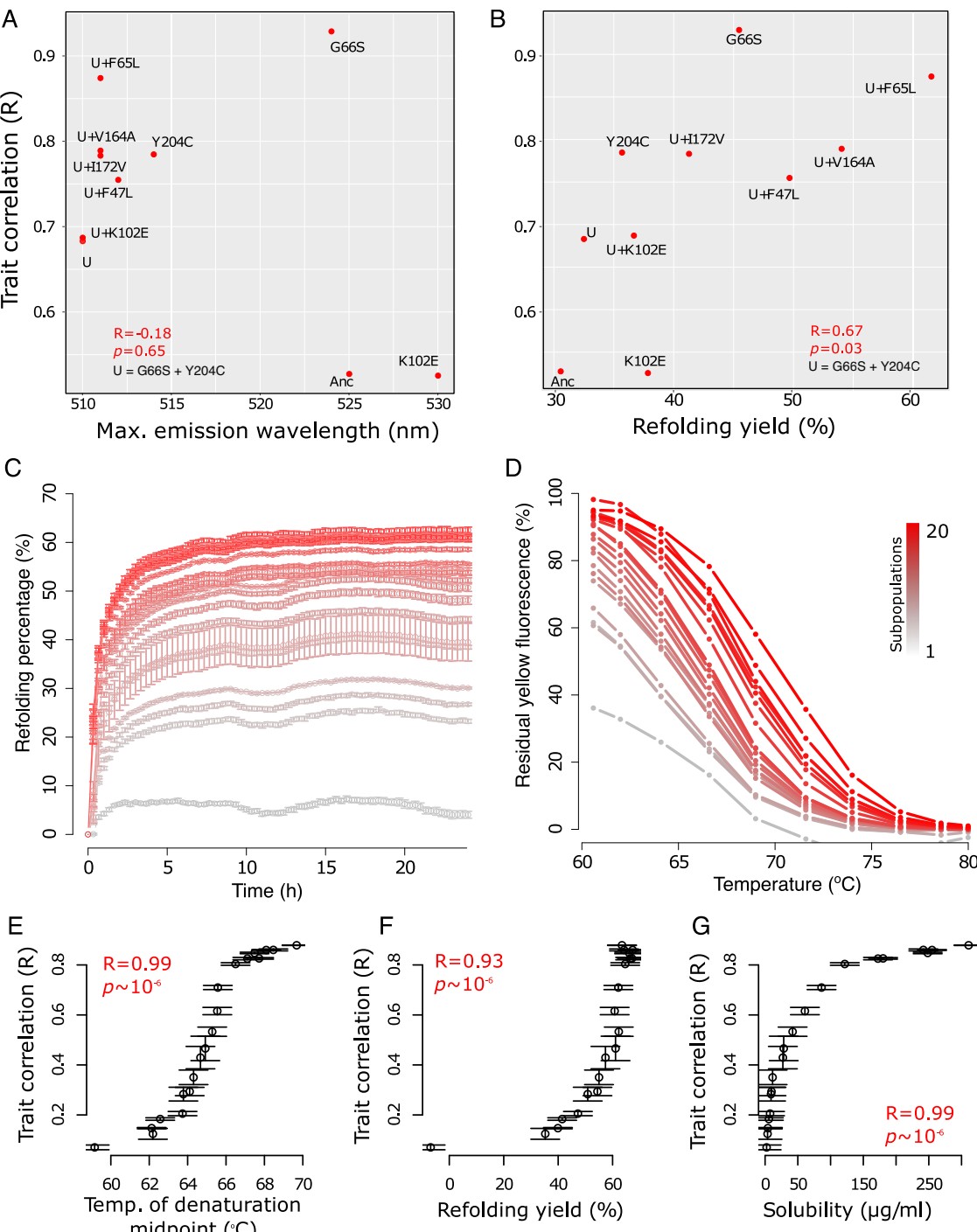

**Fig. 3 | Trait correlation substantially changes by single-point mutations that affect protein folding.** Spearman's rank correlation coefficient between yellow and green fluorescence intensities versus (**A**) maximum emission wavelength (nm) for a selected set of 10 mutants, and (**B**) refolding yield after thermal denaturation for the same mutants. **C** Refolding percentage upon thermal denaturation for the 20 subpopulations versus time (in hours). **D** Residual yellow fluorescence as a function of temperature, for the 20 subpopulations. In panels A and B, sub-populations are color coded from gray (subpopulation 1) to red (subpopulation 20). Spearman's rank correlation coefficient between yellow and green fluorescence intensities versus (**E**) the temperature of the denaturation midpoint (*Tm*) of YFP subpopulations ($p = 6.08 \times 10^{-6}$), (**F**) the percentage of refolded YFP upon thermal denaturation ($p = 6.45 \times 10^{-6}$), and (**G**) the soluble fraction of YFP (in μg/ml) as assessed by an enzyme-linked immunosorbent assay (ELISA) ($p = 6.14 \times 10^{-6}$). In panels **A** and **B**, the symbol U represents the genotype G66S-Y204C. All correlation coefficients were calculated using two-sided Spearman's rank correlations. In panels **C**, **E**–**G**, we have $n = 3$ biologically independent samples per subpopulation, and data is shown as mean values ± SD. Source data for all panels are available in the Source Data file.

To this end, we isolated up to 100 VIM-2 variants from each round of evolution and measured their MICs of the three beta-lactam antibiotics (see Methods). We observed that the correlation between the resistance (MIC) of enzyme variants to ampicillin *vs.* meropenem and ampicillin *vs.* cefotaxime varied substantially during the experiment, and significantly weakened from the first to the last round. In round one, the Spearman's rank correlation between ampicillin and cefotaxime resistance was 0.91, but by round 100, it had significantly decreased to 0.47 (Fig. 4D, Fisher transformation of Spearman's rank correlation $p < 0.001$). Likewise, the correlation between ampicillin

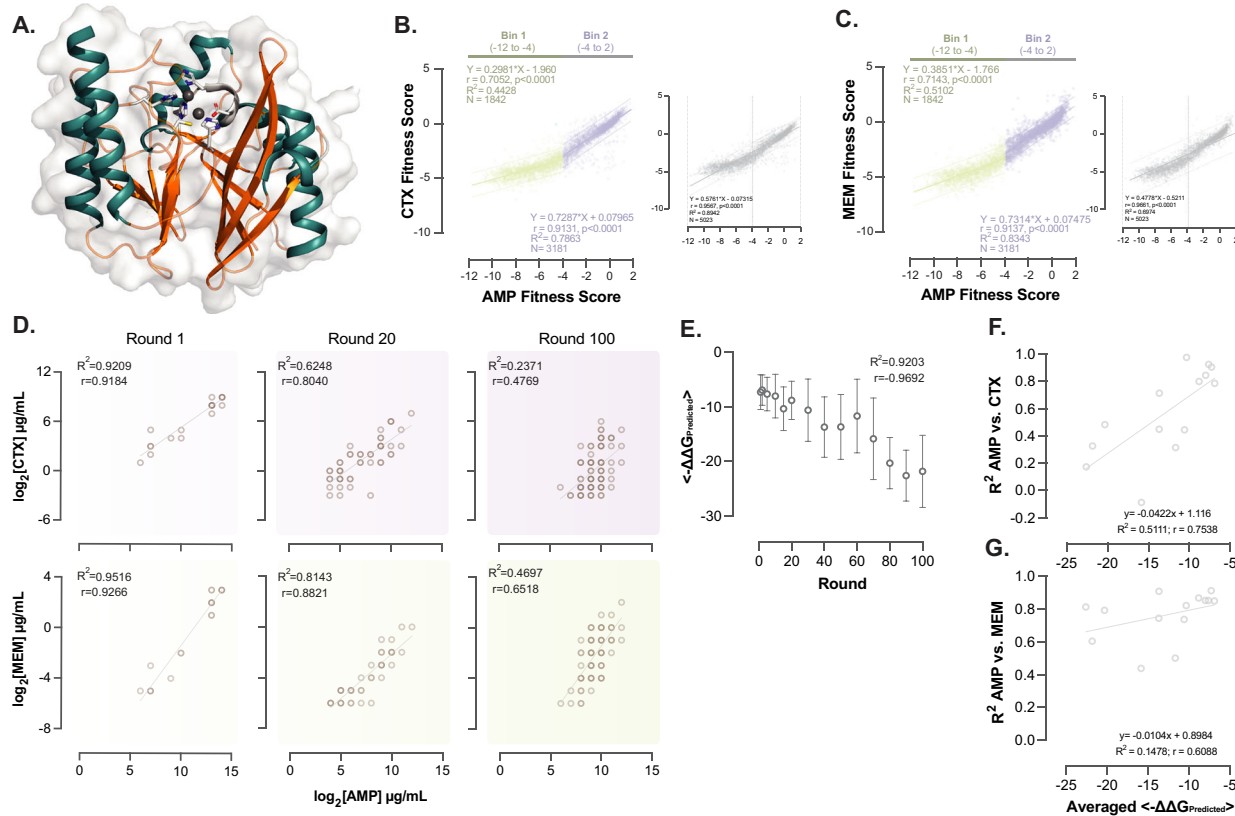

**Fig. 4 | Exploring trait correlations in an antibiotic resistance enzyme. A** The structure of wild-type metallo-beta-lactamase VIM-2 (PDB: 4bz3), depicted as a ribbon cartoon (green and orange) overlaid with the solvent accessible surface (Connolly's molecular surface, shown in gray). The side chains of active site residues are shown with the ball and stick representation alongside the two Zinc atoms (gray spheres within the protein's active site) that coordinate the hydroxyl radical responsible for the nucleophilic attack on the beta-lactam ring[27,29]. **B**, **C** Relative fitness effects of individual amino acid substitutions on VIM-2 expressed in *E. coli*, tested in media containing ampicillin vs. cefotaxime (panel B) or meropenem (panel C). The 95% prediction bands of the best-fit line are depicted as dotted lines, fitted to all data points (left panel) and when data is grouped into two fitness effect categories. Bin 1: fitness scores −12 to −4, shown in green; bin 2: fitness scores −4 to 2, shown in lilac. Gray data panels to the right of each colored plot show trait correlation for all mutations. **D** Trait correlation between resistance against ampicillin vs. cefotaxime (upper panel) and meropenem (lower panel), measured as the minimum inhibitory concentration (MIC) for representative VIM-2 populations across 100 rounds of evolution. Each circle corresponds to the data from one VIM-2 mutant. The number of mutants is $n = 24$, 48, and 93 for rounds 1, 20, and 100, respectively. **E** Predicted change in folding stability (ΔΔG) of VIM-2 variants across different rounds of experimental evolution. The number of biologically independent variants in the rounds 1, 2, 5, 10, 15, 20, 30, 40, 50, 60, 70, 80, 90, and 100 are $n = 24$, 24, 21, 21, 23, 22, 25, 29, 18, 37, 22, 23, 20 and 23, respectively. Data is shown as mean values ± SD. **F**, **G** The relationship between the average predicted folding stability of VIM-2 variants at different rounds of evolution (as shown in panel **E**) is plotted against the correlation between ampicillin and cefotaxime resistance (panel **F**), and between ampicillin and meropenem resistance (panel **G**). Source data for panels **B**–**G** are available in the Source Data file.

and meropenem resistance significantly decreased from $R = 0.92$ in round 1 to $R = 0.65$ in round 100 (Fisher transformation of Spearman's rank correlation $p < 0.001$). These findings, highlight the evolvability of trait correlations in an evolving VIM-2 population on the short time scale of laboratory evolution. In addition, throughout the evolution experiment, the correlation between ampicillin and cefotaxime resistance varied more substantially (from $R = 0.92$ in round 1 to $R = 0.47$ in round 100) than between ampicillin and meropenem resistance (from $R = 0.92$ in round 1 to $R = 0.65$ in round 100), indicating that the malleability of trait correlations might differ from one trait pair to another.

Finally, we examined how trait correlations in our antibiotic resistance enzyme may be affected by the same factor – protein foldability – that affects trait correlations in fluorescent proteins (Fig. 3). To find out, we quantified the statistical association between (i) the change in predicted folding stability of VIM-2 variants in our evolving VIM-2 population at different times during evolution and (ii) the correlation (Spearman's R) in resistance for our pairs of antibiotics. We employed this partly computational approach because the measurement of protein foldability in VIM-2 populations is not as

straightforward as in YFP, where a fluorescence assay suffices to quantify the refolding yield after thermal or chemical denaturation[32,36]. A similar measurement on VIM-2 would require enzymatic assays on purified VIM-2 proteins, which are difficult to perform for many variants sampled from protein populations. We thus sequenced ~20–30 VIM2 variants at each of multiple time points during experimental evolution, measured their antibiotic resistance, and predicted the folding stability of each variant using FoldX, a widely used predictor of this stability (see Methods; Source data). We found that the folding stability of VIM2 populations decreased during evolution (Fig. 4E). Notably, the average predicted folding stability significantly decreased with diminishing trait correlation only for ampicillin and cefotaxime resistance ($R = 0.71$ between folding stability and resistance correlation, $p \sim 0.004$; Spearman's rank correlation). This decrease is not driven by decreasing resistance itself (Supplementary Note 2). Folding stability decreased but not significantly so for ampicillin and meropenem resistance ($R = 0.36$, $p \sim 0.1$; Spearman's rank correlation). The small sample size of ~20–30 variants per population may be one cause of this observation. Another may be the more complex relationship

between trait correlation and protein folding in VIM-2 compared to YFP that we discuss below.

## Discussion

Our experiments with a fluorescent protein (YFP) and an enzyme (VIM-2) demonstrate that correlations between traits of such proteins can change dramatically and rapidly on short evolutionary time scales by the dual forces of mutation and selection. For both fluorescent proteins and for VIM-2, particularly for resistance against ampicillin and cefotaxime, we also show that trait correlations increase with protein foldability.

Foldability is essential for the function of most proteins[37–40]. It can also readily change through mutations. Specifically, most random mutations in proteins are destabilizing[41], but ~20% of such mutations are stabilizing and increase protein foldability[41]. In consequence, protein foldability is highly variable during protein evolution[42–44]. For instance, although mammalian myoglobins exhibit similar oxygen binding abilities (oxygenation constant ~ 0.8–1.2 μM − 1), their unfolding resistance to chemical denaturants differs dramatically, with up to a 600-fold variation[44,45]. More generally, proteins in the proteomes of *E. coli*, *C. elegans*, *S. cerevisiae*, and human vary widely in their thermodynamic stability[46,47], which correlates with protein foldability[34]. These examples suggest that changes in protein foldability occur frequently during protein evolution, potentially leading to rapid alterations in trait correlations. Our findings expand upon the established concept that highly stable and foldable proteins are more evolvable[1,48,49]. It is not solely the increased stability or foldability of proteins that promotes evolvability. Instead, the dynamic nature of this property can render trait correlations malleable, thereby fostering evolvability depending on whether an increased or decreased trait correlation is advantageous.

The extent to which mutations can change trait correlation is not the same for each trait, and it may depend on the number of mutations that affect the trait. For example, we observed that the correlation between resistance against meropenem and ampicillin is less malleable in evolution than the correlation between resistance to cefotaxime and meropenem (Fig. 4). A possible explanation comes from the number of amino acid positions whose mutations confer resistance to the three beta-lactam antibiotics. Specifically, mutations at 25 amino acid positions of VIM-2 alter resistance against at least one of our three antibiotics, but this number is not the same for different antibiotics. That is, mutations in 21 out of 25 positions affect ampicillin resistance, while many fewer (10 of 25) affect cefotaxime resistance, and only one affects meropenem resistance[29]. We speculate that the larger number of mutations affecting either cefotaxime or ampicillin resistance also contributes to their more malleable trait correlation. More generally, we anticipate that a protein's structure, particularly the number of amino acid positions affecting an enzyme's active site, or impacting protein function through long-range effects such as allostery, will be crucial in determining the malleability of trait correlations in proteins.

Our observations on the changes in trait correlation in protein populations help us gain more insights on the mutational pleiotropy. Most mutations are pleiotropic, that is, they simultaneously affect multiple protein traits[50–52]. A pleiotropic mutation may alter all traits to a similar extent – its effects on traits may be isotropic – or it may affect some traits differently from others. Our observations show that mutations may differ in their isotropic effect on protein traits. For example, we had previously observed that 29 single-point mutations in 25 amino acid positions within the active site of VIM-2 lead to increased resistance against all three antibiotics[29]. These mutations are isotropic in the sense that they increase resistance to multiple antibiotics. We quantified the prevalence of such mutations among 4565 mutations in 240 amino acid positions in VIM-2 using our previous deep mutational scan of VIM-2 (Supplementary Note 3, Supplementary Table 4). Approximately 2% of mutations enhanced resistance to all three antibiotics. Conversely, roughly 61% of mutations decreased resistance to the three antibiotics. Hence, in VIM-2, around 63% of all mutations are isotropic, either enhancing or diminishing resistance to all three antibiotics we studied here (Supplementary Table 4). Isotropy also varied among different amino acid positions (Supplementary Fig. 6). While we were unable to estimate the fraction of isotropic mutations in YFP due to the absence of a comprehensive mutational scan for this protein, we did examine the isotropy of 21 single-point mutants in YFP (Supplementary Table 5). Out of these 21 mutants, 12 significantly altered YFP's yellow and green fluorescence intensities relative to the wild-type. Specifically, five mutations increased and five other mutations decreased both yellow and green fluorescence. The remaining two mutations (G66S and Y204C), were anisotropic. They enhanced green fluorescence but reduced yellow fluorescence, i.e., they shifted the fluorescence color from green to yellow. Overall, our observations on both VIM-2 and YFP indicate that isotropy can vary among mutations. Systematic exploration of the pleiotropy of mutations using deep mutational scanning data may help to generalize our findings to other proteins.

Lastly, our results provide insights into the malleability of G and M matrices, two fundamental concepts in quantitative genetics[11–13]. The G matrix encapsulates the genetic variances and covariances of traits across individuals in a given population. The rapid changes in trait correlations we observe indicate that the G matrix is highly malleable in both of our study systems. The M matrix characterizes trait variances and covariances that result from mutations. One difficulty in estimating it is that most observable organismal traits are not just influenced by mutation but by mutation and selection. To determine if mutations alone can impact the M matrix, we must examine how mutations in various genetic backgrounds affect trait correlations. In YFP, we had previously created mutagenized libraries of 21 single-point mutation variants of YFP, using error-prone PCR with ~0.84 amino acid-changing mutations per YFP molecule[32]. We employed this dataset to explore how trait correlation varies across different genetic backgrounds. If this background had no effect on the M matrix, trait correlations should be identical for these YFP variants. However, contrary to this expectation, we observed significant variation in trait correlations among the mutagenized YFP variants (Supplementary Fig. 7). This observation demonstrates that the impact of mutations on trait correlations can change substantially with genetic background. It provides evidence for the malleability of the M matrix. And because the 21 variants arose in the course of a short laboratory selection experiment[32], it also suggests that selection can rapidly change the M matrix. This analysis further demonstrates that tractable molecular systems, such as the protein populations examined in this study, can help to explore fundamental questions in quantitative genetics.

It is important to acknowledge two key limitations of our study. Firstly, the pairs of protein traits we studied here are similar to each other. Although our selection assays independently targeted these traits, their inherent similarity may bias our conclusions, which may not apply to more dissimilar traits. Future research could explore the evolution of trait correlations in more dissimilar traits, such as enzymatic reactions catalyzed by a bifunctional enzyme or traits related to protein oligomeric states, such as the formation of dimers and tetramers. Secondly, we relied on predicted stabilities for different variants within VIM-2 populations. While stability predictors are widely used to assess protein stability[44,53–55], deviations from experimentally measured stabilities might affect our conclusions regarding the degree of trait correlation and its malleability. Future experiments utilizing population-level assays, such as differential scanning fluorimetry[56], may provide a better assessment of how trait correlation varies with changes in the stability or foldability of proteins.

In sum, our experiments not only show that correlations between traits of single proteins can change rapidly and on short evolutionary time scales. They also provide a simple biophysical explanation for this

change. They link the fundamental protein property of foldability with a fundamental aspect of evolvability – trait correlations. In doing so, they can help to explain how evolvability can evolve rapidly.

## Methods

### Plasmids, strains, and mutant libraries

We used the vector pBAD202/D-TOPO (K4202-01, Invitrogen), which carries an arabinose-inducible araBAD promoter and a Kanamycin-resistance gene for YFP evolution within the *E. coli* strain BW27783 (CGSC 12119). As described previously[3], we inserted the coding region of *yfp* from water jellyfish (*Aequorea victoria;* Uniprot ID: A0A059PIR9) that was already present in the plasmid pAND[2] into the vector backbone of pBAD202/D-TOPO by restriction digest and ligation. We inserted the YFP-coding gene between *Xho*I and *Hind*III restriction sites, placing it under the control of the arabinose-inducible *araBAD* promoter. We named the resulting plasmid pBAD-YFP.

We introduced random mutations into the coding region of YFP by mutagenic PCR as previously described[3]. Briefly, we added 10 ng of template plasmid to 100 µl of a PCR reaction mix that contains 10 µl of 10 × ThermoPol buffer (M0267L, NEB), 2.5 µl of *Taq* DNA polymerase (M0267L, NEB), 400 µM of dNTPs (R0192, Thermo Scientific), 3 µM of 8-oxo-GTP/dPTP (Trilink Biotechnologies), and 400 nM of each primer (MutafpF- GAAGGAGATATACctcgag /MutafpR- AGACCGTTTAAA-Caagctt). We used a Biometra thermocycler to perform PCR by using the following program: 95 °C/30 s; 25 cycles of 94 °C/20 s, 46 °C/30 s and 68 °C/50 s; 68 °C/5 min. We used the restriction enzymes *Xho*I and *Hind*III (R0146L/R3104S, NEB) to digest the resulting PCR products, and used *Dpn*I (R0176S, NEB) to remove the template plasmid by following the manufacturer's protocols. We used the QIAquick PCR purification kit to purify the digested mutated YFP pools to obtain linearized inserts.

### Sorting cells from a pool of evolving populations at the end of directed evolution

To systematically study how selection affects protein evolvability through direct effects on fitness as well as through indirect effects on protein stability and foldability, we sampled diverse YFP variants that vary broadly in their yellow fluorescence. Specifically, we sampled from YFP populations created in a previously published directed evolution experiment, in which we had evolved populations of YFP under either strong selection for yellow fluorescence (populations S, top 20 percent of yellow fluorescing cells survive selection), weak selection (populations W, top 65 percent survive), or no selection (populations N for neutral, 100 percent survive)[32]. For each of these selection regimes we had evolved four replicate populations for four rounds ("generations") of directed evolution through fluorescent activated cell sorting (FACS)-based selection and PCR-based mutagenesis.

To create a pool of yellow fluorescent proteins that cover a broad range of fluorescence intensities, we sampled 100 µl of glycerol stock from each of the four replicate S populations (from generation 4), W populations (generation 4), and W populations (generation 2). We added each of these 12 (= 4 + 4 + 4) samples into 2 ml LB medium supplemented with 30 µg/ml kanamycin. We grew the resulting 12 cultures at 37 °C with shaking at 220 rpm in a 10 ml tube for ~5 h, and then transferred 50 µl of each culture to 2 ml LB supplemented with 30 µg/ml kanamycin. After continuing cultivation for another ~12 h, we combined 400 µl of each culture into one tube and mixed thoroughly. Subsequently, we added 900 µl of the resulting mixture into 600 µl 50% glycerol and stored it at −80 °C for subsequent sorting experiments. We call the resulting mixture our "pooled" population of YFP-expressing cells. We sorted this pooled populations into 20 "subpopulations" as shown in Fig. 2A.

To sort the pooled population into 20 subpopulations according to their fluorescence, we first added 200 µl of glycerol stock of the

pooled population sample to 3 ml LB medium containing 30 µg/ml kanamycin, and grew the resulting culture at 37 °C with shaking at 220 rpm for ~5 h. We then transferred 200 µl of the culture into 20 ml LB medium with 50 µg/ml kanamycin, and continued the incubation for ~12 h. We then sampled 2 ml of the culture and centrifuged it at 9000 g and 4 °C for 5 min to collect cells. We suspended the collected cells in 2 ml LB medium supplemented with 50 µg/ml of kanamycin and 0.2% arabinose, and continued cultivation for ~12 h. Subsequently, we sampled 40 µl of culture and suspended it in 2 ml cold PBS buffer. We selected cells by their yellow fluorescence intensity according to the selection criteria described in Fig. 2A, B with an Aria III cell sorter (BD Biosciences). Specifically, we sorted cells at 4 °C in the FITC channel ($\lambda ex = 488$ nm, $\lambda em = 530 \pm 15$ nm), and collected $10^5$ cells in ~1 ml LB medium for each sorted subpopulation. We placed the selected cells on ice until we had finished sorting all subpopulations to prevent cell proliferation or death. We regrew the sorted cells and followed the same procedure to perform a second sorting for each of the twenty subpopulations, according to the selection criteria described in Supplementary Fig. 8. We prepared a glycerol stock of each subpopulation for later flow cytometry measurements.

### Fluorescence assay using flow cytometry

We added 200 µl of glycerol stock from each subpopulation into 10 ml LB medium containing 50 µg/ml of kanamycin, and incubated the resulting 20 cultures at 37 °C with shaking at 220 rpm for ~8 h. We collected cells by sampling three separate 2 ml of each subpopulation's culture and then centrifuging them at 9000 g and 4 °C to collect cells. To the pelleted cells of each subpopulation we added 2.3 ml LB medium supplemented with 50 µg/ml of kanamycin and 0.2% arabinose, resuspended the cells, and grew the resulting culture at 37 °C with shaking at 220 rpm for ~12 h. This procedure yielded 20 overnight cultures. We then added 20 µl of these 20 overnight cultures to 180 µl of cold PBS buffer. After mixing thoroughly by pipetting, we transferred 5 µl of the resulting suspension into 195 µl of cold PBS buffer, mixed thoroughly, and measured yellow fluorescence in the FITC channel ($\lambda_{ex} = 488$ nm and $\lambda_{em} = 530 \pm 15$ nm) and green fluorescence in the AmCyan channel ($\lambda_{ex} = 405$ nm and $\lambda_{em} = 525 \pm 25$ nm) at room temperature. We used a Fortessa cell analyzer (BD Biosciences) to analyze ~$10^4$ events per biological replicate with a flow rate of ~3000 events/s.

### Flow cytometry data analysis

We used FlowJo V10.4.2 (LLC) to analyze flow cytometry data. Specifically, we first selected a homogenous cell population by forward scatter height (FSC-H) versus side scatter height (SSC-H) density plots. We then selected singlets (single cells) by using side scatter area (SSC-A) versus side scatter height (SSC-H) density plots. We used the resulting filtered data for determining green and yellow fluorescence intensities.

### Extracting soluble fluorescent proteins

After inducing the expression of YFP variants in each subpopulation as described in *Fluorescence assay using flow cytometry*, we sampled 2 ml of the overnight culture of each subpopulation for extracting soluble fluorescent proteins by following the manufacturer's protocol. Specifically, we centrifuged the culture for each subpopulation at 5000 g and at 4 °C for 5 min to collect cells, and stored the collected cells at −20 °C overnight. We then followed the manufacturer's protocol to extract soluble proteins by using CelLytic™ B Cell Lysis Reagent (B7435-500ml, Sigma). Subsequently, we used 200 µl cell lysis solution (ThermoFisher; 50 mM Tri with pH 7.4, 250 mM NaCl, 5 mM, 50 mM NaF, 1 mM Na₃VO₄, 0.02% NaN3) to dissolve the soluble proteins, and placed the resulting soluble pellet sample on ice for subsequent experiments.

## Protein refolding assay

To unfold fluorescent proteins, we mixed 5 μl of crude lysate of each subpopulation with 45 μl of 8 M urea (containing 10 mM DTT), and heated the sample at 95 °C for 5 min in a PCR thermocycler. As a control, we mixed 5 μl of crude lysate with 45 μl of TNG buffer (100 mM Tris, 100 mM NaCl, 10% glycerol, 10 mM DTT, 1 × cOmplete™ (EDTA-free Protease Inhibitor Cocktail, Roche 11873580001), pH 7.2–7.5). To refold the unfolded fluorescent proteins, we rapidly added 10 μl aliquots of an unfolded sample or of the control into 190 μl of TNG buffer in a 96-well microplate, and immediately measured fluorescence intensity using an Infinite F200 Pro microplate reader ($\lambda ex$ = 485 nm, $\lambda em$ = 530 nm). We measured fluorescence at ~20 min intervals with 2-mm orbital shaking in between. We report the refolding yields as fluorescence relative to the control.

## Protein thermal stability assay

To quantify the thermal stability of fluorescent proteins in each subpopulation, we added 2 μl of crude lysate to 98 μl of TNG buffer, and mixed thoroughly by pipetting. We incubated the resulting mixture in a PCR cycler for 5 min, and subjected each subpopulation to a temperature range of 60.6–80.6 °C (specifically, 60.6, 62, 64.1, 66.6, 69, 71.6, 74, 76.5, 78.6, 80 and 80.6 °C), followed by a 30 s incubation at 4 °C. Then we immediately transferred 90 μl of each mixture to a 96-well microplate, and used an Infinite F200 Pro ($\lambda ex$ = 485 nm, $\lambda em$ = 530 nm) to measure its fluorescence intensity. As a control, we used the unheated lysate-buffer mixture. We report thermal stability as fluorescence relative to the control.

## Quantification of soluble fluorescent proteins by ELISA

To quantify the amount of soluble fluorescent proteins in each subpopulation, we used a GFP ELISA Kit (AKR-121, Cell Biolabs Inc.) which can detect GFP, BFP, CFP, and YFP from *Aequorea victoria*. Specifically, we followed the manufacture's protocol to determine the quantity of soluble fluorescent proteins in the lysate of each biological replicate for every subpopulation by comparing its absorbance with that of a recombinant GFP standard curve.

## Fluorescence assay using a microplate reader

To further validate how different selection strengths affect the association between green and yellow fluorescence, we randomly sampled 90 clones from each of the twenty subpopulations, and measured their green and yellow fluorescence intensities using a microplate reader. Specifically, we used saline to dilute the glycerol stock of each subpopulation $10^5$-fold, and plated 100 μl of the resulting culture on LB agar supplemented with 25 μg/ml kanamycin. We incubated the LB agar plates in an incubator at 37 °C overnight, picked single colonies, and inoculated each colony into 200 μl of LB medium (50 μg/ml kanamycin). We grew the resulting cultures in a microplate incubator at 37 °C and 1000 rpm. After ~5 h of incubation, we transferred 50 μl of each culture into 150 μl LB medium supplemented with 0.2% (w/v) arabinose and 50 μg/ml kanamycin, and continued the incubation for ~16 h. We then mixed 50 μl of each culture with 170 μl PBS buffer by pipetting thoroughly, and used a TECAN microplate reader (TECAN Spark) to measure both yellow ($\lambda ex$ = 485 nm, $\lambda em$ = 530 nm) and green ($\lambda ex$ = 400 nm, $\lambda em$ = 512 nm) fluorescence intensities.

## Antibiotic resistance experiments with VIM-2 metallo-beta-lactamase to obtain the fitness effects of single amino acid mutations

For our analysis we used fitness values of VIM-2 single-point mutants from our previous deep mutational scanning experiments[35]. Briefly, in these experiments we grew VIM-2 variants in the absence of antibiotics and in the presence of different concentrations of various beta-lactam antibiotics, namely 128, 16 and 2.0 μg/mL for ampicillin, 4.0 and 0.5 μg/mL for cefotaxime, and 0.031 μg/mL for meropenem. After selection,

we isolated the plasmids expressing these variants, amplified them by PCR, and sequenced the amplicons on the Illumina NextSeq 550 platform. We then calculated the fitness score of each variant by dividing the read count of that variant at each antibiotic concentration by the read count in the absence of antibiotics. We normalized these scores by dividing them by the same ratio determined for WT VIM-2.

## Long-term evolution experiments with VIM2

For our long-term experimental evolution of VIM-2, we started from the wild-type VIM-2-coding gene cloned into a low-copy number in-house plasmid with a constitutive, low expression TEM promoter and chloramphenicol resistance, which we named pIDR[29]. We generated randomly mutagenized libraries of wild-type (WT) VIM-2 via error-prone PCR (epPCR) by adding the nucleotide analogues 8-oxo-2'-deoxyguanosine-5'-triphosphate (8-oxo-dGTP) or 2'-deoxy-P-nucleoside-5'-triphosphate (dPTP) (TriLink). Each of the two 25 μL PCR reactions consisted of 1 x GoTaq Buffer (Promega), 3 μM MgCl2, 0.1 μM of each primer, 0.2 mM of dNTPs, 1.00 U of GoTaq DNA polymerase (Promega), 1 ng of template plasmid, and either 100 μM of 8-oxo-dGTP **or** 1 μM of dPTP. We programmed the first PCR (error-prone PCR) as follows: an initial denaturation step (95 °C for 2 min), followed by 20 cycles of 95 °C for 30 s, 58 °C for 60 s, 72 °C for 60 s, before a final extension step (72 °C for 3 min). We subsequently purified the PCR products with the EZ.N.A.® Cycle Pure PCR Purification Kit (OMEGA Bio-tek Inc), quantified them using a NanoDrop spectrophotometer, and used them in equal parts in a subsequent PCR reaction to ensure a balance between transition versus transversion nucleotide mutations, and a specific mutation rate and increased product yield for downstream processing. The second PCR reaction used the following reagents: 5 μL of 1 ng/ μL dPTP-epPCR product, 5 μL of 1 ng/ μL 8-oxo-dGTP-epPCR product, 1 x GoTaq Buffer (Promega), 3 μM MgCl2, 0.1 μM of each primer, 0.25 mM of dNTPs, 1.0 U of GoTaq DNA polymerase (Promega) in a final volume of 50 μL. We performed the second PCR using the same program as the first, but with 35 instead of 20 amplification cycles. We then purified the PCR products with the EZNA Cycle Pure PCR Purification Kit, and digested them with NcoI (FastDigest, ThermoFisher Scientific™) and XhoI (FastDigest, ThermoFisher Scientific™) for 1 h at 37 °C. In addition, we digested the pIDR[29] plasmid with NcoI and XhoI, for 3 h at 37 °C. Subsequently, we purified the digested plasmid from a 1% agarose gel using gel purification columns, while we purified the digested PCR products of the mutagenized VIM-2 gene with an E.Z.N.A.® Cycle Pure PCR Purification Kit. We ligated the digested VIM-2 gene fragments with the vector using a ligation mixture (10 μL) consisting of 1 × T4 DNA ligase buffer (ThermoFisher Scientific™), 5 U of T4 DNA ligase (ThermoFisher Scientific™), 8–10 ng of prepared vector, and 30–40 ng of prepared mutagenized insert. We incubated this mixture at room temperature for 3 h. We then purified the resulting ligation products with a MicroElute kit (OMEGA Bio-tek Inc.) and eluted them with 20 μL of water.

## Selection of libraries in the presence of ampicillin

we performed two consecutive evolution experiments with VIM-2[35]. In the first, we evolved WT VIM-2 towards higher ampicillin resistance for 18 rounds of evolution, until it had reached a plateau of ~ 40-times higher resistance than WT VIM-2. The second experiment started from the surviving population at the end of the first experiment and lasted for an additional 100 rounds. During this time, we kept the selection pressure constant at 10 μg/mL ampicillin.

For both experiments, we first transformed 4–5 μL of the purified pIDR/VIM-2 ligation products into E.cloni® 10 G E. coli cells (Lucigen Corp.) using electroporation. We performed the first directed evolution experiment, where we iteratively mutagenized and selected the resulting VIM-2 variants in increasing ampicillin concentrations until they had reached a resistance plateau, as follows. We grew E. coli cells transformed with the mutagenized VIM-2 pool overnight at 30 °C in

10 mL of LB media supplemented with 34 µg/mL of chloramphenicol. Then we plated 100 µL of a 1:100 dilution of the overnight culture onto a series of LB agar plates containing increasing ampicillin concentrations, with a 2-fold concentration increase from plate to plate, which ranged from 2 to 8,192 µg/mL. We used colonies from the plate with the highest concentration of ampicillin and colony counts between 100 and 1000 colonies for the next experimental step. For this step, we scraped all colonies and extracted the plasmids to use them as templates for the next round of mutagenesis. We repeated this process of selection and mutagenesis iteratively for 18 rounds, where we increased the ampicillin concentration used for selection from 256 µg/mL to 4096 µg/mL during the first 9 rounds, and kept this concentration for the rest of the evolution at 4096 µg/mL, because the number of surviving colonies dropped below our lower threshold for the next concentration (8192 µg/mL). For the second, long-term evolution experiment, we used the population obtained at the end of the first directed evolution experiment as our starting point and repeated the same iterative mutagenesis and selection cycle using large LB agar plates and ampicillin as the selection pressure, but kept the ampicillin concentration on the selective plates constant at 10 µg/mL. We repeated this process iteratively for 100 rounds. We supplemented all plates with 34 µg/mL chloramphenicol (Cm) to select for the successful uptake and expression of the plasmid.

### Measuring the minimum inhibitory concentration of individual VIM-2 variants

To quantify the ampicillin resistance level conferred on *E. coli* by individual variants isolated from VIM2 libraries, we used agar-plate based assays to determine the minimum inhibitory concentration (MIC) of antibiotics for *E. coli* carrying a specific VIM-2 variant. To this end, we grew *E. coli* cells harboring a single VIM2 variant in 500 µL LB-Cm media at 30 °C overnight in a deep-96-well plate. The next day, we inoculated 5 µL of the overnight culture into 195 µL of LB-Cm in quadruplicate in a standard 96-well plate, and grew the resulting cultures for 3 h at 37 °C. We then plated the cultures with 96-well replicator pins on a series of 15 mm LB agar plates harboring increasing levels of antibiotics (two-fold increases in ampicillin, meropenem, and cefotaxime from 2 to 32,768 µg/mL, 0.016 to 64 µg/mL, and 0.032 to 4096 µg/mL respectively). We incubated these agar plates overnight at 37 °C. The next day, we determined the MIC of each variant by identifying the concentration of antibiotics at which no growth was observed in at least three of the four replicates for each variant.

### Sequencing of individual variants and calculation of ΔΔG

We randomly picked 24–96 single colonies from selected libraries, and PCR amplified the VIM-2 gene region of the pIDR plasmid with NEB Taq2x Master Mix using the manufacturer's protocol, with an initial denaturation (95 °C for 2 min), followed by 30 cycles of amplification (95 °C for 30 s, 58 °C for 60 s, and 72 °C for 60 s), before a final extension step (72 °C for 3 min). We subsequently purified the resulting PCR products enzymatically by treatment with ExoI (ThermoFisherScientific™) and FastAP (ThermoFisherScientific™) for 1 h at 37 °C, and then inactivated the enzymes via heat treatment of the sample by incubation at 85 °C for 15 min. We sent the purified products for Sanger sequencing (Azenta™). We visually inspected sequencing results in Geneious® bioinformatics software and identified mutations by comparing each mutant VIM-2 gene sequence to the wild-type sequence using an in-house Python script. We used FoldX to estimate the change in free energy of folding (ΔΔG) for all sequenced mutants in our libraries, using the crystal structure of wtVIM2 (PDB ID=4bz3).

### Statistical analyses

To test the null hypothesis that trait correlations are identical between samples (e.g., when comparing trait correlations between two sub-populations), we used Fisher's z-transformation. In this statistical method, Pearson's or Spearman's correlation coefficients are converted to z-scores, so that they become normally distributed. The null hypothesis is then tested using a t-test on the z-scores. We performed all data and statistical analyses with R (v4.2.1)[57], Python (v3.10.4), PRISM™ 9.0 software, SMRT Link (v9.0.0.92188), and FlowJo (v10.4.2 & 10.8.1).

### Reporting summary

Further information on research design is available in the Nature Portfolio Reporting Summary linked to this article.

## Data availability

The experimental data used to generate all figures are available in the source file. All the FACS files for YFP populations are available at the following GitHub link: https://github.com/dasmeh/Trait_correlation. We used the 3D structure of VIM2 with the pdb ID=4BZ3 (https://www.rcsb.org/structure/4BZ3) for structure visualization. We used previously published deep mutational scanning data of VIM2 for trait correlation analyses performed on mutational data (raw data available at: BioProject ID PRJNA606894, processed data available at: https://cdn.elifesciences.org/articles/56707/elife-56707-supp2-v2.xlsx).
Source data are available as a Source Data file. Source data are provided with this paper.

## Code availability

Scripts and statistical analyses are available at the GitHub repository: https://github.com/dasmeh/Trait_correlation.

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

## Acknowledgements

We would like to acknowledge the center for human genetics of Marburg University and federal ministry of education and research in Germany (PD, PerMED-COPD program), European Research Council (AW, grant agreement no. 739874), Swiss National Science Foundation (AW, grant 31003A_172887), Westlake Education Foundation (JZ), the National Natural Science Foundation of China (JZ, grant numner: 32270669), and Canadian Institute of Health Research (CIHR) (NT, project grants: AWD-019305 and AWD-018386), for financial supports. The project on which

this report is based was funded by the Federal Ministry of Education, and research under the funding code 01EK2203A. Responsibility for the content of this publication is up to the author.

## Author contributions

Conceptualization and study design: P.D., J.Z., A.W., A.N.E. and N.T. Analysis: P.D., J.Z. and A.N.E. Manuscript writing: P.D., J.Z., A.W., A.N.E., and N.T.

## Competing interests

The authors declare no competing interests.
