## [Peer Review File · Nature Communications]

Rapid evolutionary change in trait correlations of single proteinsReviewers' Comments:

Reviewer #1:

Remarks to the Author:

This was an interesting read and an impressive laboratory experimental study. The authors use some clever methods to investigate the lability of trait correlations using the *E. coli*-system and two colours as "traits" (yellow and green fluorescence) which are governed by the same underlying protein. The study is elegant and the results are strong and seemingly convincing. The study is also well-anchored conceptually and theoretically, and relevant literature is cited.

The laboratory methods are far beyond my expertise, having no experience of the *E. coli*-system, so I will mainly focus on some conceptual points below, that could hopefully be useful to the authors when revising the manuscript. My comments are hopefully perceived as constructive and not overly critical.

1. I struggle a little bit over the definition of "traits" in this study. The use of two different colours (yellow and green) seems quite operational to me (it is simply something that is possible to measure), but conceptually I am not entirely convinced that these are two independent traits. Rather, one could argue it is a single trait (=colour) and since none of the two colours seem to have any direct effect on fitness they are essentially neutral, right? Perhaps the authors could develop their rationale and motivate their choice of these two traits a little bit more in depth, even though I understand this is a tricky issue as we get in to the more philosophical domain of the character concept in evolutionary biology, and what counts as a character etc.

2. The use of two colour traits that are neutral to fitness reminds me of some models of sexual selection and sympatric speciation, when an "arbitrary" trait (e. g. colour) becomes genetically correlated to a fitness-related traits under selection with ecological function(e. g. body size) and linkage disequilibrium develops. However, in this particular *E. coli*-system, the authors are actually using a simpler system, as it is a single protein which affects two different "traits", i. e. a case of pleiotropy and a gene affecting multiple traits. In that sense the results are maybe not as surprising as they would have been if the two colours were governed by different genes and proteins and the genetic correlation would change due to increased linkage disequilibrium, which would probably be more challenging to achieve. Would it even be possible to do such a complementary experiment in this system?

3. Related to 2 above: The results in the present study shows - to my understanding - that there might be both standing genetic variation in the degree of pleiotropy and novel mutations also vary in their degree of pleiotropy, both which are interesting findings. Perhaps the authors could be explicit about this and briefly discuss it?

4. The issue of of pleiotropy of novel mutations have been discussed recently, and there are some different opinions. Some biologists have argued that the naive null hypothesis in quantitative genetics is isotropy, i. e. that novel mutations have phenotypic effects that are more or less equal in all directions (Uller et al. (2018). *Genetics* 209: 949-966). This is disputed by others, who instead emphasize that there is no such assumption in quantitative genetics, and that, on the contrary, mutational pleiotropy is the norm, and not the exception (Svensson & Berger (2019). *Trends Ecol. Evol.* 34: 422-434). It seems to me that the results in the present study point more to novel mutations frequently being pleiotropic, rather than isotropic, right?

5. Other examples of mutational pleiotropy are discussed in Svensson et al (2021; cited already cited by the authors as Ref. 4), but could be mentioned and maybe briefly discussed as well, e. g.:

Reddiex AJ, Chenoweth SF (2021). *Proc R Soc B: Biol Sci* 288:20211785.

McGuigan K, Collet JM, Allen SL, et al (2014). *Genetics* 197:1051-1062.

<https://doi.org/10.1534/genetics.114.165720>

McGuigan K, Collet JM, McGraw EA, et al (2014). *Genetics* 196:911-921.
<https://doi.org/10.1534/genetics.114.161232>

6. I realize that this is not a quantitative-genetic study, but given that the authors want to connect their single-protein, single-locus study to the broader evolutionary literature, I wonder if it would be worth briefly connecting their results to the literature on G- and M-matrix evolution and the relationship between standing genetic variation as captured by G and novel genetic variation as captured by M? What is the relationship between G and M, and can their interesting results have any bearing on this question? How would we expect G and M to be correlated to each other, if at all? And how would such a correlation arise, mechanistically and evolutionarily? This is discussed in Svensson et al. (2020) and Svensson & Berger (2019), which cite the following papers in the quantitative genetic literature, which might also optionally be referred to here:

Houle et al. (2017). *Nature* 548(7668):447-450:

These authors showed strong concordance between G, M and the divergence matrix between species over 40 million years of flywing evolution, raising the question of how such similarity could arise and be maintained.

Jones et al. (2014). *Nature Communications* 5: 1-10:

These authors used simulations to show that we would expect triple alignment between G, M and the selective surface, implying that correlational selection shapes G, but also indirectly M and thereby evolvability. I think such a link between standing genetic variation and novel mutations is principally important in the evolvability discussion that also the present study on *E. coli* alludes to. If the authors of the present study can connect their results to this broad literature on phenotypic traits and quantitative genetics, I think they will do a great service to the evolutionary biology research community at large, so it might be worth some effort.

7. I struggle a little bit to understand the type of selection regime the authors employed, although I might have missed this. Did they select on one trait (e. g. yellow GFP) and then greenness was dragged along a correlated response? This would only explain why greenness also changed - as a simple correlated response - but could hardly explain why the correlation was strengthened? Or did they select directly on protein folding capacity, which would make more sense, and could mechanistically explain the strengthened correlation between the two colour traits? Maybe you explain it in the article, but forgive me for not being an expert on *E. coli* and microbiology experiments of this kind.

8. Incidentally, would it technically be possible to achieve a negative genetic correlation between yellow and green? Given that there is variation in the degree of pleiotropy among novel mutations, it should at least be theoretically possible, or not? If it would be possible, have the authors considered this? If it is not possible, wouldn't that maybe indicate that the two different "traits" could not alternatively be considered as a single traits, measured in different ways?

I congratulate the authors to an elegant and clever study, and hope that my questions were not too naive and have been useful when they revise their manuscript. I waive my anonymity, should the authors want to contact me or ask for clarifications.

Lund (Sweden) May 31 2022

Erik Svensson
(erik.svensson@biol.lu.se)

Reviewer #2:

Remarks to the Author:

This manuscript uses measurements of two fluorescent traits across a number of strains and populations to study the evolution of trait correlations. The Introduction lays out a well-written and concise case for why we care about trait correlations and their evolution. I think there is merit in attempting to address these issues with single-gene systems. However, I have some concerns about the analysis and interpretation of these data, and how relevant this study can really be to the sort of systems and questions invoked in this Introduction.

At various points, I think this manuscript is not specific enough about what kind of correlations are being referred to and compared. For example, the central comparison in fig. 1, between C and D, contrasts a correlation caused by stochastic differences to one caused by stochastic + genetic differences. The finding that these are different is not good evidence for the claim that mutations can change this baseline correlation, if "baseline correlation" refers to the correlation caused by stochastic differences alone. This issue with the approach is similar to Simpson's paradox, in that a correlation among the means of different subgroups is being substituted for the correlation of data within a subgroup. Now, eyeballing that data I think it is likely that these mutations are changing the correlations between the traits, but to prove this either specific strains need to be measured over multiple individuals, or some more sophisticated model needs to be fit to these data.

These issues continue in the next section, which looks at correlations in populations with an unspecified degree of genetic variation. Subdividing the cells by yellow fluorescence does provide more resolution, but doesn't partition the variation between genotype and stochastic effects. Without that partitioning, I don't see how differences between populations can be meaningfully interpreted. Fig. 2G&H does present analyses by clone—I think these data should be the focal point of this comparison.

The data in fig. 3 are much more compelling because they avoid this ambiguity. However, I am still a little concerned by the choice of strain for panel C and what it implies. Based on this one example, it seems like the correlation disappears when the protein is essentially not fluorescing, because autofluorescent contributions to these measurements lack a correlation. It seems, then, that the degree of positive correlation is likely to vary with the magnitude of fluorescence contributed by yfp rather than the cell background, and Fig. 3E as well as subsequent arguments in the paper support this interpretation. Is this as interesting and generalizable as presented in this manuscript? I am not too convinced. If we view autofluorescence as purely experimental noise, rather than part of the traits of interest per se, then the evolving correlations seem likely to be spurious—i.e., they would disappear if not for autofluorescence. Perhaps, though, we should take the traits as including this background effect of autofluorescence. In that case, I am still not convinced that the changes in correlations observed here are of general interest to an audience interested in evolving trait correlations. In either case, I think there is way too much analysis here relative to the simplicity of the point being communicated—that the quantity of active protein relative to autofluorescent signal scales the correlation. This issue is, for me, compounded when we consider that Fig. 1, and to some extent Fig. 2, present less cleanly interpretable versions of the same patterns contained in fig. 3. I think this paper might be more welcome in a more concise presentation, focusing on the strongest lines of evidence.

Circling back to Fig. 1, I remain confused why the mutagenized distribution of fluorescence looks the way it does. The center of the wild-type distribution is about (2^{15} , $2^{8.25}$), but not a single one of the mutants is near that point (though some are higher). Also, many of the mutants are below the mean background level of autofluorescence for the yellow measurement—in fact, they are below the confidence interval for it, at least by eye. I suspect that these experiments are very hard to do reproducibly, and that the authors are focusing on correlations in part because the absolute

measurements are not easy to compare across different experiments. If that's the case, it certainly isn't a fatal issue. However, I think these discrepancies should be noted and discussed in the manuscript.

Based on these issues, I certainly think that the manuscript could be productively revised, but that it might benefit from a pretty major overhaul.

Notes

Lines 75-76: I think this is a little optimistic, as it ignores some issues of scale. Trait correlations in polygenic traits might be very malleable over the long term but not change very much after a single substitution, precisely because many genes influence the correlation. You have made a sufficient case for investigating correlations at the level of a single gene without making this leap.

Lines 79-80: Again, this syllogism isn't really valid to me, because we know that yfp has expression costs, often considerable ones, even though it doesn't perform a native function per se.

Line 83: Pedantically, FACS isn't necessary for phenotyping, only for the additional element of selecting, right?

Line 95: This section seems out-of-place—maybe a header here?

Fig. 1: I would favor keeping the axes ranges the same across panels here, unless it majorly obscures information.

Line 141-142: I would imagine that cell-size variation would also be a significant factor here, right?

Lines 159-162: This sentence is very hard to parse—suggest putting the ratios in parentheses following the treatments. Also, are these ratios flipped—presumably, strong selection divided by no selection is the 18-fold comparison?

Reviewer #3:

Remarks to the Author:

In this manuscript, Dasmeh et al attempted to study how evolution affects two correlated phenotypes. The authors chose YFP as the model system and yellow and green fluorescence as the correlated properties. The manuscript builds on their recent paper (ref 30) where YFP evolution landscape for green and yellow emissions under varied selection pressure was thoroughly characterized. By analyzing a mutant YFP library by FACS, they show that the correlation between the intensities of yellow and green emission got stronger with a more stringent selection for enhanced yellow fluorescence. A deeper look into the mutation landscape revealed that stabilizing mutations augment both green and yellow fluorescence. The work is technically sound and the manuscript is written well.

A deeper understanding of how evolution affects related phenotypes in the same system would indeed be valuable, as the authors discuss at the outset. However, the model chosen by the authors is too simple a system to offer a broader understanding of this grand question. Yellow and green fluorescence coming out of YFP variants are not even too distinct properties. Both are emanating from the same chromophore with the same absorption and emission spectra. Measuring yellow and green fluorescence intensities of a YFP mutant is kind of like measuring the height of the same Gaussian hill at two different vantage points. Not surprisingly, mutations that increase the absorbance/fluorescence overall would positively impact both, and enhance the correlation. It is also possible that there are shifts in the absorption and emission maxima, selectively favoring one or the other, which would be more interesting. But these cannot be differentiated because the authors simply characterize the green and yellow fluorescence intensities of each mutant, instead of the entire absorption and

emission spectrum. Providing the latter characterization, and in a way that enables quantitative comparison, would likely provide a better understanding of what really drives the correlation. In the end, I really wish the authors chose a more sophisticated system with parameters that are correlated but not almost the same thing (e.g., enzyme activity and stability), that would have provided a more meaningful insight. The key conclusion from this study – the impact of stabilizing mutants on enhancing correlated properties – makes sense, but the novelty of the conclusion is weakened by the similar observations the authors have reported in their recent papers including ref 30.

Overall, this paper attempts to tackle an important question and reports a large body of careful data, but enthusiasm is significantly dampened by the overly simplistic model and the similarity of the key conclusion to other recent studies.

Reviewer 1

This was an interesting read and an impressive laboratory experimental study. The authors use some clever methods to investigate the lability of trait correlations using the *E. coli*-system and two colours as "traits" (yellow and green fluorescence) which are governed by the same underlying protein. The study is elegant and the results are strong and seemingly convincing. The study is also well-anchored conceptually and theoretically, and relevant literature is cited. The laboratory methods are far beyond my expertise, having no experience of the *E. coli*-system, so I will mainly focus on some conceptual points below, that could hopefully be useful to the authors when revising the manuscript. My comments are hopefully perceived as constructive and not overly critical.

Authors: Before we respond point-by-point, we would like to thank the reviewer very much for the constructive comments and suggestions. They indeed helped us to connect our results with a broader literature on evolvability, and improve the manuscript substantially.

1. I struggle a little bit over the definition of "traits" in this study. The use of two different colours (yellow and green) seems quite operational to me (it is simply something that is possible to measure), but conceptually I am not entirely convinced that these are two independent traits. Rather, one could argue it is a single trait (=colour) and since none of the two colours seem to have any direct effect on fitness they are essentially neutral, right? Perhaps the authors could develop their rationale and motivate their choice of these two traits a little bit more in depth, even though I understand this is a tricky issue as we get in to the more philosophical domain of the character concept in evolutionary biology, and what counts as a character etc.

Authors: We agree that the yellow and green colors, which we define as two traits, are different from traits that are more directly linked to fitness. However, we treat these colors as two traits for the following reasons. First and most importantly, we can evolve fluorescent proteins towards emitting either of the two colors, which can help us understand how evolution of one color can affect the evolution of the other color. This question is analogous to one that is asked in fields such as quantitative genetics, about how the evolution of one morphological trait affects another trait. Second, our two traits can be easily quantified with high throughput assays such as FACS, and the underlying genetic causes are also easy to determine. Third, although the two colors have no direct effect on fitness (cell growth), because we select mutants based on their fluorescence, fluorescence determines survival and can serve as a proxy for fitness. Just as important, fluorescence activated cell sorting allows selection of individual cells based on this fitness proxy, which much greater control of the selection pressure compared to other experimental evolution approaches.

The choice of fluorescent proteins in general has two additional benefits for our analyses. First, being non-native to *E. coli*, they interact minimally with the rest of the *E. coli* proteome and thus minimize biases and confounding factors that might otherwise affect our observations. Second, they help us to reliably measure molecular properties of proteins such as protein stability because using fluorescence assays are less challenging than assays requiring protein purification.

That being said, this comment (and comments of other reviewers) motivated us to also expand this paper to a system whose traits are more directly linked to cellular fitness. For this purpose, we chose the VIM-2 metallo- β -lactamase (MBL), a highly effective enzyme that can hydrolyze different classes of beta-lactam antibiotics, allowing bacterial cells to survive in the presence of these antibiotics. In this study system, we consider resistance to the three different antibiotics of ampicillin, cefotaxime, and meropenem as our traits and examine how selection on one trait (ampicillin) leads to changes in trait correlations. These additional experiments confirm our assertion that trait correlations can rapidly change by both mutation and selection.

Here is the list of changes we made to the manuscript (highlighted in yellow) to discuss our new system:

We revised the abstract (lines 15-29):

“Many organismal traits are genetically determined and covary in evolving populations. The resulting trait correlations can either help or hinder evolvability – the ability to bring forth new and adaptive phenotypes. The evolution of evolvability requires that trait correlations themselves must be able to evolve, but we know little about this ability. To learn more about it, we here study two evolvable systems, a yellow fluorescent protein and the antibiotic resistance protein VIM2 metallo beta-lactamase. We consider two traits in the fluorescent protein, namely the ability to emit yellow and green light, and three traits in our enzyme, namely the resistance against ampicillin, cefotaxime, and meropenem. We show that correlations between these traits can evolve rapidly through both mutation and selection on short evolutionary time scales. In addition, we show that these correlations are driven by a protein’s ability to fold, because single mutations that alter foldability can dramatically change trait correlations. Since foldability is important for most proteins and their traits, mutations affecting protein folding may alter trait correlations mediated by many other proteins. Thus, mutations that affect protein foldability may also help shape the correlations of complex traits that are affected by hundreds of proteins.”

We added the following paragraph to introduction to introduce VIM2 (lines 87-97):

“The second gene extends our analysis to traits that are directly linked to cellular fitness. It is the gene *vim2*, which encodes the protein VIM2 metallo betalactamase (M β L). VIM-2 metallo- β -lactamase is a highly effective enzyme that confers broad-spectrum resistance against beta-lactam antibiotics. VIM-2 belongs to the genetically and functionally diverse M β L superfamily, which has the remarkable ability to efficiently hydrolyze distinct classes of β -lactam antibiotics²⁷⁻²⁹. Given its broad-spectrum resistance capabilities, VIM-2 is an ideal candidate for experiments on trait correlations. The VIM-2 traits that we study are resistance to the three different antibiotics

ampicillin, cefotaxime, and meropenem. We have previously shown that only a few mutations suffice to change the resistance conferred by VIM2 to these antibiotics^{27,30}. Here we quantify correlations between these antibiotic resistance traits.”

Our manuscript has now a whole new section where we discuss our observations on VIM2 (lines 296-417).

We also compared trait correlation for different traits of VIM2 (lines 436-451):

“The extent to which mutations can change trait correlation is not the same for each trait, and it may depend on the number of mutations that affect the trait. For example, we observed that the correlation between resistance against meropenem and ampicillin is less malleable in evolution than the correlation between resistance to cefotaxime and meropenem (Figure 4). A possible explanation comes from the number of amino acid positions whose mutations confer resistance to the three beta-lactam antibiotics. Specifically, mutations at 25 amino acid positions of VIM2 alter resistance against at least one of our three antibiotics, but this number is not the same for different antibiotics. That is, mutations in 21 out of 25 positions affect ampicillin resistance, while many fewer (10 of 25) affect cefotaxime resistance, and only one affects meropenem resistance²⁹. We speculate that the larger number of mutations affecting either cefotaxime or ampicillin resistance also contributes to their more malleable trait correlation. More generally, we anticipate that a protein's structure, particularly the number of amino acid positions affecting an enzyme's active site, or impacting protein function through long-range effects such as allostery, will be crucial in determining the malleability of trait correlations in proteins.”

2. The use of two colour traits that are neutral to fitness reminds me of some models of sexual selection and sympatric speciation, when an "arbitrary" trait (e. g. colour) becomes genetically correlated to a fitness-related traits under selection with ecological function(e. g. body size) and linkage disequilibrium develops. However, in this particular E. coli-system, the authors are actually using a simpler system, as it is a single protein which affects two different "traits", i. e. a case of pleiotropy and a gene affecting multiple traits. In that sense the results are maybe not as surprising as they would have been if the two colours were governed by different genes and proteins and the genetic correlation would change due to increased linkage disequilibrium, which would probably be more challenging to achieve. Would it even be possible to do such a complementary experiment in this system?

Authors: This is an interesting thought! It is possible in principle to implement such an experiment using our current system. To do so, one could evolve two linked fluorescent proteins with one emitting strong yellow but weak green fluorescence and another emitting weak yellow but strong green fluorescence.

However, because these genes would have similar sequence (and thus can lead to uncontrolled recombination), and because mutagenesis should not break up the linkage between them, these experiments would be much more challenging than the experiments discussed in this manuscript. Indeed, this could be an interesting (multi-year) project for future investigations.

3. Related to 2 above: The results in the present study shows - to my understanding - that there might be both standing genetic variation in the degree of pleiotropy and novel mutations also vary in their degree of pleiotropy, both which are interesting findings. Perhaps the authors could be explicit about this and briefly discuss it? The issue of pleiotropy of novel mutations have been discussed recently, and there are some different opinions. Some biologists have argued that the naive null hypothesis in quantitative genetics is isotropy, i. e. that novel mutations have phenotypic effects that are more or less equal in all directions (Uller et al. (2018). *Genetics* 209: 949-966). This is disputed by others, who instead emphasize that there is no such assumption in quantitative genetics, and that, on the contrary, mutational pleiotropy is the norm, and not the exception (Svensson & Berger (2019). *Trends Ecol. Evol.* 34: 422-434). It seems to me that the results in the present study point more to novel mutations frequently being pleiotropic, rather than isotropic, right? Other examples of mutational pleiotropy are discussed in Svensson et al (2021; cited already cited by the authors as Ref. 4), but could be mentioned and maybe briefly discussed as well, e. g.:

Reddiex AJ, Chenoweth SF (2021). *Proc R Soc B: Biol Sci* 288:20211785.

McGuigan K, Collet JM, Allen SL, et al (2014). *Genetics* 197:1051-1062.

<https://doi.org/10.1534/genetics.114.165720>

McGuigan K, Collet JM, McGraw EA, et al (2014). *Genetics* 196:911-921.
<https://doi.org/10.1534/genetics.114.161232>

Authors: We fully agree! Our trait correlation analyses for mutants and evolving populations support the notion that mutations are pleiotropic. They further show that mutations differ in their isotropic effects. We added the following paragraph to Discussion and cited the suggested literature (lines 452-477):

Our observations on the changes in trait correlation in protein populations help us gain more insights on the mutational pleiotropy. Most mutations are pleiotropic, that is, they simultaneously affect multiple protein traits⁴⁹⁻⁵¹. A pleiotropic mutation may alter all traits to a similar extent – its effects on traits may be isotropic – or it may affect some traits differently from others. Our observations show that mutations may differ in their isotropic effect on protein traits. For example, we had previously observed that 29 single-point mutations in 25 amino acid positions within the active site of VIM2 lead to increased resistance against all three antibiotics²⁹. These mutations are isotropic in the sense that they increase resistance to multiple antibiotics.

We quantified the prevalence of such mutations among 4565 mutations in 240 amino acid positions in VIM2 using our previous deep mutational scan of VIM2 (Supplementary Note 3, Table S4). Approximately 2% of mutations enhanced resistance to all three antibiotics. Conversely, roughly 61% of mutations decreased resistance to the three antibiotics. Hence, in VIM2, around 63% of all mutations are isotropic, either enhancing or diminishing resistance to all three antibiotics we studied here (Table S4). Isotropy also varied among different amino acid positions (Figure S6). While we were unable to estimate the fraction of isotropic mutations in YFP due to the absence of a comprehensive mutational scan for this protein, we did examine the isotropy of 21 single point mutants in YFP (Table S5). Out of these 21 mutants, 12 significantly altered YFP's yellow and green fluorescence intensities relative to the wild-type. Specifically, five mutations increased and five other mutations decreased both yellow and green fluorescence. The remaining two mutations (G66S and Y204C), were anisotropic. They enhanced green fluorescence but reduced yellow fluorescence, i.e., they shifted the fluorescence color from green to yellow. Overall, our observations on both VIM2 and YFP indicate that isotropy can vary among mutations. Systematic exploration of the pleiotropy of mutations using deep mutational scanning data may help to generalize our findings to other proteins.

We added Figures S5, Figure S6, Table S4, and Table S5 to the Supplementary Information and cited the suggested literature in references 49-54.

We also like to briefly discuss the reviewers' interesting thought on the pleiotropic nature of novel mutations. While our results generally indicate isotropic effects for most mutations, we have noted that mutations at amino acid positions crucial for the evolution of novel functions tend to exhibit anisotropic effects. As previously mentioned in our manuscript, the mutations G66S and Y204C in YFP, resulting in a color shift from yellow to green, exemplify this anisotropic effect. Our investigation of VIM2 further supports this observation, revealing significantly lower trait correlations for mutations within the active site, implying a greater degree of anisotropy (see Figure R1 below). This insight aligns with the reviewer's assertion that early-arising novel mutations, particularly those impacting critical functional residues, are likely more anisotropic than mutations occurring later in evolution. We are now considering the preparation of a concise paper to delve deeper into this concept and appreciate the reviewer's valuable insight to this intriguing problem.

Figure R1. Trait correlation for mutations in the active site amino acids compared with mutations in the rest of the protein. A) The fitness scores of VIM2 variants under cefotaxime resistance with those under ampicillin resistance specifically for the active site amino acids. B) The fitness scores of VIM2 variants under cefotaxime resistance with those under ampicillin resistance for the remaining protein sequence. C) The fitness scores of VIM2 variants under meropenem resistance with those under ampicillin resistance, for the active site amino acids. D) The fitness scores of VIM2 variants under meropenem resistance with those under ampicillin resistance for the rest of the protein sequence. The concentration of ampicillin, cefotaxime, and meropenem in these experiments were 128 $\mu\text{g/ml}$, 4 $\mu\text{g/ml}$, and 0.031 $\mu\text{g/ml}$, respectively. All correlation coefficients presented are Spearman's correlation coefficients. The active site amino acids considered are located at positions 39, 55, 57, 63, 117, 119, 142, 143, 153, 196, 219, 222, 243, 248, 62, 67, 68, 60, 61, 202, 205, 210, 211, 216, and 201, with reference to the VIM2 structure (PDB: 5yd7).

4. I realize that this is not a quantitative-genetic study, but given that the authors want to connect their single-protein, single-locus study to the broader evolutionary literature, I wonder if it would be worth briefly connecting their results to the literature on G- and M-matrix evolution and the relationship between standing genetic variation as captured by G and novel genetic variation as captured by M? What is the

relationship between G and M, and can their interesting results have any bearing on this question? How would we expect G and M to be correlated to each other, if at all? And how would such a correlation arise, mechanistically and evolutionarily? This is discussed in Svensson et al. (2020) and Svensson & Berger (2019), which cite the following papers in the quantitative genetic literature, which might also optionally be referred to here:

Houle et al. (2017). *Nature* 548(7668):447-450:

These authors showed strong concordance between G, M and the divergence matrix between species over 40 million years of fly wing evolution, raising the question of how such similarity could arise and be maintained.

Jones et al. (2014). *Nature Communications* 5: 1-10:

These authors used simulations to show that we would expect triple alignment between G, M and the selective surface, implying that correlational selection shapes G, but also indirectly M and thereby evolvability. I think such a link between standing genetic variation and novel mutations is principally important in the evolvability discussion that also the present study on *E. coli* alludes to. If the authors of the present study can connect their results to this broad literature on phenotypic traits and quantitative genetics, I think they will do a great service to the evolutionary biology research community at large, so it might be worth some effort.

Authors: We greatly appreciate the reviewer's suggestion. It enriched the scope of our manuscript and gave us new ideas for possible future experiments.

We agree that our results have relevance for the stability and evolution of G and M matrices in quantitative genetics. They suggest that the G matrix in our system is malleable and can change by mutations and selection over a few of generations. However, to estimate the M matrix one would need to establish a protein sequence-based predictive model that provides a direct linkage between genotype and phenotype. Such a model could be used to estimate the variance-covariance matrix of phenotypic caused by mutations in each generation. This is beyond the scope of the current manuscript. Nonetheless, our findings provide evidence for the malleability of the M matrix. We examined the trait correlation in mutagenized libraries of YFP which we had previously generated and found that trait correlation significantly changes for such libraries.

We added the following paragraph to Discussion to elaborate on this point (lines 478-498):

Lastly, our results provide insights into the malleability of G and M matrices, two fundamental concepts in quantitative genetics^{4,11-13,49,55}. The G matrix encapsulates the genetic variances and covariances of traits across individuals in a given population. The rapid changes in trait correlations we observe indicate that the G matrix is highly malleable in both YFP and VIM2, in line with previous observations in other systems^{56,57}. The M matrix characterizes trait variances and covariances that result from mutations. One difficulty in estimating it is that most observable organismal traits are not just influenced by mutation but by mutation and selection. To determine if mutations alone can impact the M matrix, we must examine how mutations in various genetic backgrounds affect trait correlations. In YFP, we had previously created mutagenized libraries of

21 single point mutation variants of YFP, using error-prone PCR with ~0.84 amino acid-changing mutations per YFP molecule³². We employed this dataset to explore how trait correlation varies across different genetic backgrounds. If this background had no effect on the M matrix, trait correlations should be identical for these YFP variants. However, contrary to this expectation, we observed significant variation in trait correlations among the mutagenized YFP variants (Figure S7). This observation demonstrates that the impact of mutations on trait correlations can change substantially with genetic background. It provides evidence for the malleability of the M matrix. And because the 21 variants arose in the course of a short laboratory selection experiment³², it also suggests that selection can rapidly change the M matrix. This analysis further demonstrates that tractable molecular systems, such as the protein populations examined in this study, can help to explore fundamental questions in quantitative genetics.

We added Table S7 to the Supplementary Information and cited the suggested literature in references 49, 55-57.

5. I struggle a little bit to understand the type of selection regime the authors employed, although I might have missed this. Did they select on one trait (e. g. yellow GFP) and then greenness was dragged along a correlated response? This would only explain why greenness also changed - as a simple correlated response - but could hardly explain why the correlation was strengthened? Or did they select directly on protein folding capacity, which would make more sense, and could mechanistically explain the strengthened correlation between the two colour traits? Maybe you explain it in the article, but forgive me for not being an expert on E. coli and microbiology experiments of this kind.

Authors: Thank you for raising this question, which allowed us to eliminate an ambiguity in the text. We selected directly for high yellow fluorescence (and that indirectly led to more efficient protein folding). We now state this more explicitly on lines 159-163:

We next addressed our second focal question: Can selection change trait correlations? To answer it, we measured the correlation between yellow and green fluorescence intensities in YFP populations that we had previously evolved under multiple cycles of mutation and either strong selection, weak selection, or no selection **for yellow fluorescence intensity**^{32,33} (Figure 2A, see Methods for details).

6. Incidentally, would it technically be possible to achieve a negative genetic correlation between yellow and green? Given that there is variation in the degree of pleiotropy among novel mutations, it should at least be theoretically possible, or not? If it would be possible, have the authors considered this? If it is not possible, wouldn't that maybe indicate that the two different "traits" could not alternatively be considered as a single traits, measured in different ways?

Authors: We did observe a negative genetic correlation between yellow and green fluorescent for some mutants. For example, the G66S and Y204C mutations can shift the emission spectrum from yellow to green, leading to a strong enhancement of green fluorescence but a strong decrease of yellow fluorescence (Table S5, Figure S5). We also observed that in VIM2 metallo beta lactamase, our new study system, ~43% of mutations have anisotropic effects on different traits which includes mutants with a negative trait correlation as well as those that lack a significant one. We also hope that our new experiment with VIM2 and the resistance traits addresses the concern that trait correlation should be observed between two distinct traits. We also elaborated on this point in the revised version (lines 135-146), which now reads:

We next turned to our first focal question: Can mutations strengthen or weaken this baseline correlation. To answer this question, we measured trait correlations for 71 YFP mutants that we had previously engineered, because they attained moderate to high frequency in evolving populations^{32,33}. These variants include the WT protein, as well as 10 mutants with one, 28 mutants with two, and 32 mutants with three amino acid changes (Table S1). All double mutations share the mutation G66S (replacement of a glycine with serine at position 66 of YFP) or Y204C, and all triple mutations share both amino acid changes G66S and Y204C. **The mutations G66S and Y204C are unique in that they shift the emission spectrum of YFP from yellow towards green fluorescence³², reducing yellow fluorescence but enhancing green fluorescence. The nature of these mutations shows that yellow and green fluorescence are two distinct features of YFP and can be treated as separate (albeit possibly correlated) traits.**

I congratulate the authors to an elegant and clever study, and hope that my questions were not too naive and have been useful when they revise their manuscript. I waive my anonymity, should the authors want to contact me or ask for clarifications. Lund (Sweden) May 31 2022
Erik Svensson (erik.svensson@biol.lu.se)

Authors: Thank you again for your useful and constructive suggestions! These suggestions have helped us substantially improve the quality of the manuscript and sparked valuable discussions.

Reviewer 2

This manuscript uses measurements of two fluorescent traits across a number of strains and populations to study the evolution of trait correlations. The Introduction lays out a well-written and concise case for why we care about trait correlations and their evolution. I think there is merit in attempting to address these issues with single-gene systems. However, I have some concerns about the analysis and interpretation of these data, and how relevant this study can really be to the sort of systems and questions invoked in this Introduction.

Authors: We thank the reviewer for raising several important points which we have addressed, and which have helped us improve the manuscript. Please find the responses to each point below.

1. At various points, I think this manuscript is not specific enough about what kind of correlations are being referred to and compared. For example, the central comparison in fig. 1, between C and D, contrasts a correlation caused by stochastic differences to one caused by stochastic + genetic differences. The finding that these are different is not good evidence for the claim that mutations can change this baseline correlation, if “baseline correlation” refers to the correlation caused by stochastic differences alone. This issue with the approach is similar to Simpson’s paradox, in that a correlation among the means of different subgroups is being substituted for the correlation of data within a subgroup. Now, eyeballing that data I think it is likely that these mutations are changing the correlations between the traits, but to prove this either specific strains need to be measured over multiple individuals, or some more sophisticated model needs to be fit to these data. These issues continue in the next section, which looks at correlations in populations with an unspecified degree of genetic variation. Subdividing the cells by yellow fluorescence does provide more resolution, but doesn’t partition the variation between genotype and stochastic effects. Without that partitioning, I don’t see how differences between populations can be meaningfully interpreted. Fig. 2G&H does present analyses by clone—I think these data should be the focal point of this comparison.

Authors: Thank you for raising this important point. Since the major focus of our study is that mutations and selection can change trait correlation, we agree with the reviewer that it’s better to write about the role of mutations in changing trait correlation early in the manuscript. We thus revised our results section and, as suggested by the reviewer, started our analyses by comparing trait correlations of the WT protein to those of mutants. Particularly we made the following changes:

Our new Figure 1 (panel C) now compares the trait correlation of the WT mutant to that of mutants.

We now focus on the role of mutations earlier in the results section. Specifically, we write (lines 135-156):

We next turned to our first focal question: Can mutations strengthen or weaken this baseline correlation. To answer this question, we measured trait correlations for 71 YFP mutants that we had previously engineered, because they attained moderate to high frequency in evolving populations^{32,33}. These variants include the WT protein, as well as 10 mutants with one, 28

mutants with two, and 32 mutants with three amino acid changes (Table S1). All double mutations share the mutation G66S (replacement of a glycine with serine at position 66 of YFP) or Y204C, and all triple mutations share both amino acid changes G66S and Y204C. The mutations G66S and Y204C are unique in that they shift the emission spectrum of YFP from yellow towards green fluorescence³², reducing yellow fluorescence but enhancing green fluorescence. The nature of these mutations shows that yellow and green fluorescence are two distinct features of YFP and can be treated as separate (albeit possibly correlated) traits.

Trait correlations varied significantly among YFP mutants (Figure 1C). Specifically, they varied from $R=0.17$ (for the triple mutant G66S-Y204C-F72S; $p < 10^{-16}$, Spearman's rank correlation) to $R=0.98$ (for the double mutant G66S-N145S, $p < 10^{-16}$; Spearman's rank correlation). Remarkably, even some single point mutations sufficed to substantially strengthen or weaken the correlations of these traits relative to the WT protein. We further investigated whether changes in trait correlations arise from changes in the proportion of cells exhibiting autofluorescence or low fluorescence intensity. We found instead that they stem from variation in the fluorescence intensity of functional and actively fluorescing molecules (Supplementary note 1). Altogether, these observations show that trait correlations in our system can change substantially by single point mutations.

2. The data in fig. 3 are much more compelling because they avoid this ambiguity. However, I am still a little concerned by the choice of strain for panel C and what it implies. Based on this one example, it seems like the correlation disappears when the protein is essentially not fluorescing, because autofluorescent contributions to these measurements lack a correlation. It seems, then, that the degree of **positive correlation is likely to vary with the magnitude of fluorescence contributed by yfp rather than the cell background**, and Fig. 3E as well as subsequent arguments in the paper support this interpretation. Is this as interesting and generalizable as presented in this manuscript? I am not too convinced. If we view autofluorescence as purely experimental noise, rather than part of the traits of interest per se, then the evolving correlations seem likely to be spurious—i.e., they would disappear if not for autofluorescence. Perhaps, though, we should take the traits as including this background effect of autofluorescence. In that case, I am still not convinced that the changes in correlations observed here are of general interest to an audience interested in evolving trait correlations. In either case, I think there is way too much analysis here relative to the simplicity of the point being communicated—that the quantity of active protein relative to autofluorescent signal scales the correlation. This issue is, for me, compounded when we consider that Fig. 1, and to some extent Fig. 2, present less cleanly interpretable versions of the same patterns contained in fig. 3. I think this paper might be more welcome in a more concise presentation, focusing on the strongest lines of evidence.

Authors: This is a very good question! As we already mentioned in our response to the first point, we revised the manuscript and focused on a direct comparison of mutants earlier on. Nonetheless, we agree with the reviewer that a proper dissection of the role of auto-fluorescing cells on trait correlation is necessary. If the changes in trait correlation stem only from auto-fluorescing cells, we would expect that the variation in trait correlation is explained more significantly by variation in trait correlations among cells with a low fluorescence intensity than between cells with a high fluorescence intensity.

To find out, we calculated for each mutant the correlation between green and yellow fluorescence, doing so first for cells whose yellow fluorescence is below the median fluorescence of the population (R_{low}), and then for cells whose fluorescence is higher than the median fluorescence of the population (R_{high}). We first observed that R_{high} was significantly higher than R_{low} , showing that the trait correlation is higher for cells whose fluorescence intensity is higher. In addition, the overall trait correlation is influenced more strongly by R_{high} than by R_{low} , as inferred from a multiple linear regression model. Specifically, the estimated coefficients values of R_{high} and R_{low} were ~ 0.57 ($\text{Pr}(> |t|) \sim 10^{-13}$), and ~ 0.25 ($\text{Pr}(> |t|) \sim 10^{-7}$). We further performed this regression analysis for the cells in the 20th and 10th quantile of fluorescence. That is, we calculated R_{low} for cells whose fluorescence intensity fell below the 10th percentiles of the population's fluorescence intensity distribution, and we calculated R_{high} for cells whose fluorescence intensities were above this threshold. For cells in the 10th percentile, trait correlation is indeed not explained by R_{low} . The estimated coefficient values of R_{high} and R_{low} were ~ 0.97 ($\text{Pr}(> |t|) \sim 10^{-63}$), and ~ 0.013 ($\text{Pr}(> |t|) \sim 0.5$). This demonstrates that what changes the overall trait correlation significantly is the trait correlation in the cells with viable fluoresce intensities and not auto-fluorescing cells.

We added the following sentences to results (lines 147-156):

Trait correlations varied significantly among YFP mutants (Figure 1C). Specifically, they varied from $R=0.17$ (for the triple mutant G66S-Y204C-F72S; $p < 10^{-16}$, Spearman's rank correlation) to $R=0.98$ (for the double mutant G66S-N145S, $p < 10^{-16}$; Spearman's rank correlation). Remarkably, even some single point mutations sufficed to substantially strengthen or weaken the correlations of these traits relative to the WT protein. We further investigated whether changes in trait correlations arise from changes in the proportion of cells exhibiting autofluorescence or low fluorescence intensity. We found instead that they stem from variation in the fluorescence intensity of functional and actively fluorescing molecules (Supplementary note 1). Altogether, these observations show that trait correlations in our system can change substantially by single point mutations.

In addition, we created a new supplementary Table S3 with the results of these multiple linear regression models.

Table S3. Results of a multiple linear regression analysis for the relationship between the overall trait correlation, R , and the trait correlations between green and yellow fluorescence intensities of cells whose yellow fluorescence are surpassing the indicated quantile (R_{high}) and those falling below this quantile (R_{low}).

lm ($R \sim R_{\text{high}} + R_{\text{low}}$)					
quantile		Estimate ^a	Std. Error ^b	t-value ^c	Pr (> t) ^d
10%	(Intercept)	0.03443	0.01398	2.463	0.0163
	R_{high}	0.97906	0.02609	37.52	2×10^{-16}
	R_{low}	0.01375	0.02181	0.63	0.5306
20%	(Intercept)	0.09367	0.019	4.93	5.44×10^{-6}
	R_{high}	0.84997	0.03695	23.003	$< 2 \times 10^{-16}$
	R_{low}	0.13367	0.02825	4.732	1.14×10^{-5}
50%	(Intercept)	0.25065	0.02511	9.981	5.04×10^{-15}
	R_{high}	0.57882	0.06310	9.174	1.44×10^{-13}
	R_{low}	0.25397	0.04625	5.491	6.23×10^{-7}

a: the estimated coefficient for the predictor variable in the regression equation. b: the standard error of the coefficient estimate. c: the t-statistic, for the test of the null hypothesis that the coefficient estimate is not significantly different from zero. d: the p-value associated with the t-value. It indicates the probability of observing a t-value as extreme as the one obtained, assuming the null hypothesis is true.

We also explained our analysis in Supplementary Note 1:

In this analysis we aimed to differentiate between changes in trait correlation that arise from changes in the genetic background, i.e., they are properties of specific mutants, or simply from changes in the fraction of cells with weak or no fluorescence. Specifically, and for an isogenic population of each mutant, we calculated two correlations between the yellow and green fluorescence intensities. The first correlation was between these intensities for cells whose yellow fluorescence fell below the 50% of the overall population's fluorescence (denoted as R_{low}). The second correlation was between these intensities for cells whose yellow fluorescence fell above the 50% of the overall population's fluorescence (denoted as R_{high}). We repeated these calculations with the percentile of 20% and 10% as well. As shown in Table S3, we find that for each mutant population and for the three percentiles (10%, 20%, 50%), R_{high} has a more significant impact on the overall trait correlation than R_{low} ($p \sim 10^{-16}$; multiple linear regression models). This suggests that the changes in trait correlation among genotypes stem from variations in the fluorescence intensity of functional and actively fluorescing molecules.

3. Circling back to Fig. 1, I remain confused why the mutagenized distribution of fluorescence looks the way it does. The center of the wild-type distribution is about (2^{15} , $2^{8.25}$), but not a single one of the mutants is near that point (though some are higher). Also, many of the mutants are below the mean background level of autofluorescence for the yellow measurement—in fact, they are below the confidence interval for it, at least by eye. I suspect that these experiments are very hard to do reproducibly, and that the

authors are focusing on correlations in part because the absolute measurements are not easy to compare across different experiments. If that's the case, it certainly isn't a fatal issue. However, I think these discrepancies should be noted and discussed in the manuscript. Based on these issues, I certainly think that the manuscript could be productively revised, but that it might benefit from a pretty major overhaul.

Authors: Thank for raising this concern. The difference arose mainly because we performed fluorescence measurements in two different FACS instruments. As correctly pointed out by the reviewer we did not compare the absolute fluorescence values because of instrument-specific measurement biases and focused instead on the correlation between our traits of interest. Because this panel is not essential for our conclusions, we removed it from the figure.

Notes

Lines 75-76: I think this is a little optimistic, as it ignores some issues of scale. Trait correlations in polygenic traits might be very malleable over the long term but not change very much after a single substitution, precisely because many genes influence the correlation. You have made a sufficient case for investigating correlations at the level of a single gene without making this leap.

Authors: Thanks for this note! We removed the reference to complex traits in this passage.

Lines 79-80: Again, this syllogism isn't really valid to me, because we know that yfp has expression costs, often considerable ones, even though it doesn't perform a native function per se.

Authors: Indeed, there are inherent costs associated with *yfp* expression. Nevertheless, it's crucial to emphasize that for the specific scope of our experiments, the absence of YFP's interaction with the proteins of the host cell mitigates a source of bias in the calculation of trait correlation that might be caused by such interactors. In response to this comment, we rephrased the sentence to make it less objectionable. Specifically, we now write (lines 80-82):

"Because this protein is not native to the microbial host organism *E. coli* in which we study it, we can study its traits with less interference from the host's proteome and physiology than would be possible for native proteins."

Line 83: Pedantically, FACS isn't necessary for phenotyping, only for the additional element of selecting, right?

Authors: Entirely correct. However, in our works FACS serves the dual purpose of cell-sorting and also concurrently measuring fluorescence on a single-cell basis.

Line 95: This section seems out-of-place—maybe a header here?

Authors: Point well taken. However, we have substantially revised and shortened this segment of the manuscript (now lines 100-156), and feel that the shortened text may be better placed in the current section on the malleability of trait correlation by mutations.

Fig. 1: I would favor keeping the axes ranges the same across panels here, unless it majorly obscures information.

Authors: Thanks for this important point. The current revised Figure 1 presents only one single-cell fluorescence data set and resolves the issue raised by the reviewer.

Line 141-142: I would imagine that cell-size variation would also be a significant factor here, right?

Authors: Good point. Cell size can be quantified by measuring the FSC (Forward Scatter). In brief, FSC quantifies the intensity of light scattered in the forward direction as cells pass through the laser beam of a FACS machine. Larger cells generally exhibit higher FSC values due to increased light scattering. Importantly, cell size does not bias our results, because we focus on the correlation between green and yellow fluorescence, rather than absolute fluorescence intensities. However, we did calculate the correlation coefficient between FSC and yellow fluorescence intensity in our mutants, which ranged from ~ 0.1-0.4 for different mutants. This correlation is only weakly associated with the trait correlation (Spearman's $R = 0.23$, p -value = 0.04), further highlighting that cell size does not confound our findings.

Lines 159-162: This sentence is very hard to parse—suggest putting the ratios in parentheses following the treatments. Also, are these ratios flipped—presumably, strong selection divided by no selection is the 18-fold comparison?

Authors: Thanks for catching this problem. The ratios are correct. A YFP population under selection for yellow fluorescence has a higher yellow intensity compared to a population under no selection. However, we revised the sentence for clarity, and it now reads (lines 164-171):

“The properties of such variants differ from that of wild-type (WT) YFP. To quantify these differences, we calculated the ratio of the median yellow fluorescence of different populations to that of an isogenic population of YFP wild-type. For populations under no selection this ratio was approximately 0.68. For populations under weak selection it was approximately 1, and for populations under strong selection it was 18. Previous single-molecule real-time sequencing had also shown that during experimental evolution, YFP accumulated up to ~6 amino acid changes compared to the wild-type³².

Authors: Thanks again for the careful reading of the manuscript and for the constructive feedback.

Reviewer 3

In this manuscript, Dasmeh et al attempted to study how evolution affects two correlated phenotypes. The authors chose YFP as the model system and yellow and green fluorescence as the correlated properties. The manuscript builds on their recent paper (ref 30) where YFP evolution landscape for green and yellow emissions under varied selection pressure was thoroughly characterized. By analyzing a mutant YFP library by FACS, they show that the correlation between the intensities of yellow and green emission got stronger with a more stringent selection for enhanced yellow fluorescence. A deeper look into the mutation landscape revealed that stabilizing mutations augment both green and yellow fluorescence. The work is technically sound and the manuscript is written well. A deeper understanding of how evolution affects related phenotypes in the same system would indeed be valuable, as the authors discuss at the outset. However, the model chosen by the authors is too simple a system to offer a broader understanding of this grand question. Yellow and green fluorescence coming out of YFP variants are not even too distinct properties. Both are emanating from the same chromophore with the same absorption and emission spectra. Measuring yellow and green fluorescence intensities of a YFP mutant is kind of like measuring the height of the same Gaussian hill at two different vantage points. Not surprisingly, mutations that increase the absorbance/fluorescence overall would positively impact both, and enhance the correlation. It is also possible that there are shifts in the absorption and emission λ_{max} , selectively favoring one or the other, which would be more interesting. But these cannot be differentiated because the authors simply characterize the green and yellow fluorescence intensities of each mutant, instead of the entire absorption and emission spectrum. Providing the latter characterization, and in a way that enables quantitative comparison, would likely provide a better understanding of what really drives the correlation. In the end, I really wish the authors chose a more sophisticated system with parameters that are correlated but not almost the same thing (e.g., enzyme activity and stability), that would have provided a more meaningful insight. The key conclusion from this study – the impact of stabilizing mutants on enhancing correlated properties – makes sense, but the novelty of the conclusion is weakened by the similar observations the authors have reported in their recent papers including ref 30. Overall, this paper attempts to tackle an important question and reports a large body of careful data, but enthusiasm is significantly dampened by the overly simplistic model and the similarity of the key conclusion to other recent studies.

Authors: We really appreciate the reviewer's suggestion regarding the choice of the system. We also agree with the reviewer that the two colors of a fluorescent proteins are biologically less important than traits that are directly linked to fitness. As this issue was also raised by other reviewers, we decided to substantially expand our analysis to an enzyme whose activity is highly relevant for bacterial fitness. To this end, we collaborated with one of the world's leading protein evolution labs (the Tokuriki lab), and joined forces to investigate trait correlation in this system. This required new experiments, and new analyses of the resulting data, which are the reason why this revision took so long to prepare.

Specifically, the new analysis quantifies correlations in the resistance to the three antibiotics ampicillin, cefotaxime, and meropenem in the antibiotic conferring enzyme VIM2. We revised the manuscript, adding new sections to the results, as well as new material to the introduction, and discussion to accommodate the reviewer's concern. Briefly, our new analysis conveys the same message as our analysis of YFP. Trait

correlations are highly malleable through both mutation and selection. Here is the list of changes we made to the manuscript.

We revised the abstract (lines 15-29):

Many organismal traits are genetically determined and covary in evolving populations. The resulting trait correlations can either help or hinder evolvability – the ability to bring forth new and adaptive phenotypes. The evolution of evolvability requires that trait correlations themselves must be able to evolve, but we know little about this ability. To learn more about it, we here study two evolvable systems, a yellow fluorescent protein and the antibiotic resistance protein VIM2 metallo beta-lactamase. We consider two traits in the fluorescent protein, namely the ability to emit yellow and green light, and three traits in our enzyme, namely the resistance against ampicillin, cefotaxime, and meropenem. We show that correlations between these traits can evolve rapidly through both mutation and selection on short evolutionary time scales. In addition, we show that these correlations are driven by a protein's ability to fold, because single mutations that alter foldability can dramatically change trait correlations. Since foldability is important for most proteins and their traits, mutations affecting protein folding may alter trait correlations mediated by many other proteins. Thus, mutations that affect protein foldability may also help shape the correlations of complex traits that are affected by hundreds of proteins.

We added the following paragraph to introduction to introduce VIM2 (lines 87-97):

The second gene extends our analysis to traits that are directly linked to cellular fitness. It is the gene *vim2*, which encodes the protein VIM2 metallo beta-lactamase (MβL). VIM-2 metallo-β-lactamase is a highly effective enzyme that confers broad-spectrum resistance against beta-lactam antibiotics. VIM-2 belongs to the genetically and functionally diverse MβL superfamily, which has the remarkable ability to efficiently hydrolyze distinct classes of β-lactam antibiotics^{27,29}. Given its broad-spectrum resistance capabilities, VIM-2 is an ideal candidate for experiments on trait correlations. The VIM-2 traits that we study are resistance to the three different antibiotics ampicillin, cefotaxime, and meropenem. We have previously shown that only a few mutations suffice to change the resistance conferred by VIM2 to these antibiotics^{27,30}. Here we quantify correlations between these antibiotic resistance traits.

We added a whole new section to our manuscript and showed that trait correlations exist in VIM2 population, change by mutations, and are malleable during laboratory evolution on short time scales. (lines 296-417).

Finally, we also added a new section to the discussion to discuss the trait specificity of trait correlation in VIM2 (lines 436-451):

The extent to which mutations can change trait correlation is not the same for each trait, and it may depend on the number of mutations that affect the trait. For example, we observed that the correlation between resistance against meropenem and ampicillin is less malleable in evolution than the correlation between resistance to cefotaxime and meropenem (Figure 4). A possible explanation comes from the number of amino acid positions whose mutations confer resistance to the three beta-lactam antibiotics. Specifically, mutations at 25 amino acid positions of VIM2 alter resistance against at least one of our three antibiotics, but this number is not the same for different antibiotics. That is, mutations in 21 out of 25 positions affect ampicillin resistance, while many fewer (10 of 25) affect cefotaxime resistance, and only one affects meropenem resistance²⁹. We speculate that the larger number of mutations affecting either cefotaxime or ampicillin resistance also contributes to their more malleable trait correlation. More generally, we anticipate that a protein's structure, particularly the number of amino acid positions affecting an enzyme's active site, or impacting protein function through long-range effects such as allostery, will be crucial in determining the malleability of trait correlations in proteins.

Authors: Thanks again for the constructive feedback on choosing a different system to show trait correlation. This has helped us to extend our observations to an important antibiotic resistance conferring enzymes and discuss the generalizability of our results.

Reviewers' Comments:

Reviewer #1:

Remarks to the Author:

This is an impressive study, that became even more impressive after the revision of the original manuscript and by the addition of new data and traits (antibiotic resistance traits) that nicely confirmed what the authors already found when they only had data on the fluorescent colour traits in the previous version.

The results are very important also outside microbial ecology and evolution, as they speak to the broader research community of evolutionary biologists who are interested in evolvability and the malleability of trait correlations. I therefore appreciate that they authors included a brief paragraph at the end of their revised Discussion, where they also connected their results to the quantitative genetic literature on G- and M-matrix evolution.

From what I can see, they have carried out a study that seems very complete - at least to a non-expert on *E. coli* like me. I congratulate the authors to this impressive work, and I have learned a lot myself by reading this manuscript.

Reviewer #2:

Remarks to the Author:

I am glad to see this manuscript again, as I felt that the problems with the earlier version were very fixable. The authors have not only addressed my issues but added an entire other system. One would think that this would result in an overly bloated paper, but by simplifying the YFP analysis the paper altogether works much better.

I have a few notes below about presentation details, but none rise to the level of a significant problem. I think the paper is interesting, thoughtfully done, and that any further substantial work one might suggest would be outside the scope of a single paper. Therefore, I recommend acceptance.

Notes

The Intro ends without any indication of what you found. I think one or sentences summarizing the major results would be fitting. Overall, the introduction is good but leaves us without a very clear hypothesis-testing framework for the paper. I would say you are testing the hypothesis that trait correlations can evolve rapidly at the level of single genes. It might be productive to explicitly discuss your paper in these terms in the Intro.

Figure 2: I can't remember what is new here and what isn't, but overall this figure is good—I particularly like showing the data from the extremes in panels C-E. But, I'm puzzled by the one red dot—could you indicate the correlation expected from mutation as a line or band in this panel, so that a reader can follow this a bit more easily?

Figure 3 is low-res and barely readable. I think that some of these panels could be supplemental, as I feel this figure is a bit redundant; then again, a structural biologist (which I am certainly not) might really want all this detail in the main text.

Lines 305-310: I would rephrase this to be more specific and also more positive—you're measuring correlations across genotypes, averaging over stochastic individual differences. This is very valid, but distinguishing between these levels of correlations would help.

Lines 311-330: Maybe just say once that you're measuring Spearman correlations and collect these low-information p-values in a supplemental table?

Lines 331-346: Is this enzyme secreted at high enough levels to constitute a public good? If so, you might not be seeing mutation-selection balance per se, but a cheater-producer balance as well. I guess this may not matter for present purposes, but I was distracted by the question.

Line 408: Missing table number?

Reviewer #3:

Remarks to the Author:

In this revised manuscript, Dasmeh et al. took into account many of the comments raised by the reviewers. In particular, the inclusion of new data on VIM2 beta-lactamase evolution relieved some of the original concern about the two traits of YFP (green and yellow fluorescence) being insufficiently distinct. I think the large body of data on these two proteins over several rounds of laboratory evolution, and the analysis framework to quantify the impact of mutations on two measurable phenotypes are valuable contributions to explore the overall question of how the evolvability of distant traits is related.

Although I think that these contributions are sufficiently valuable to justify publication in Nature Communications, the results do not really live up to the expectations the manuscript set at the beginning talking about the evolvability of truly distinct traits in biological systems. This is because of two reasons: A) The traits the authors deal with here are too closely related and do not reflect the full spectrum of possible 'relatedness' of other traits in proteins such as K_m and k_{cat} , or oligomeric state and activity, or K_d for two co-substrates or a substrate and a cofactor, and numerous other potential examples that one can come up with. B) The take-home message of the paper – mutations that affect the foldability of a protein impact two very closely related phenotypes – is not novel; it is both one of the most intuitive solutions, and one that has been an established paradigm in the directed evolution community (including prior papers from the same group).

As the reviewers have noted, the traits being discussed here are very closely related, chosen because of operational convenience, rather than their truly distinctive nature. I am not persuaded by the authors' arguments on why yellow and green fluorescence are two different traits of YFP. As I noted earlier, a fluorescent protein has a single fluorescence spectrum; just because one can measure it at two different wavelengths to get two different numbers does not make these distinct traits. Different substrates for VIM are a bit more distinct, albeit they are still very closely related to the same active site and catalytic mechanism. Now, there is nothing wrong with this approach. When dealing with a complex question like this, one needs to break it down into simpler systems that are experimentally tractable. However, the limitations of dealing with such simplified systems must be acknowledged in the discussion, instead of pretending that the question has been fully addressed for good. I encourage the authors to engage in such discussion and talk about the types of studies that would be needed to more broadly explore connections between traits that are not so closely related.

Additionally, some other issues that should be addressed:

- Line 144-146: I do not agree with this assertion. Just because a mutation shifts the fluorescence spectra of a protein a few nm in either direction, does not make emissions in these wavelengths distinct traits

- Line 306-308: While not exactly single-cell measurement, plating the library at a dilution where each cell can give rise to a single colony does enable quantitative measurement of survivability of single cell(s) on an antibiotic challenge, thus providing an estimate of the activity of distinct mutants in these cells. In fact, this has been the basis of numerous directed evolution experiments.

- Line 318: "how they"

- Line 348: Table Sx?

- Line 408: Table Sx?

- Overall the data from the VIM experiments used to generate the figures should be added. I do worry a little about the stability of these mutants being theoretical instead of experimental. If there is a way to confirm that, for a subset of the mutants, the theoretical and experimental values match, that would strengthen the paper.

Reviewer 2. I am glad to see this manuscript again, as I felt that the problems with the earlier version were very fixable. The authors have not only addressed my issues but added an entire other system. One would think that this would result in an overly bloated paper, but by simplifying the YFP analysis the paper altogether works much better. I have a few notes below about presentation details, but none rise to the level of a significant problem. I think the paper is interesting, thoughtfully done, and that any further substantial work one might suggest would be outside the scope of a single paper. Therefore, I recommend acceptance.

Authors: We appreciate the reviewer's positive feedback and constructive comments on our submitted manuscript. Below, we addressed the remaining points.

Reviewer 2. The Intro ends without any indication of what you found. I think one or sentences summarizing the major results would be fitting. Overall, the introduction is good but leaves us without a very clear hypothesis-testing framework for the paper. I would say you are testing the hypothesis that trait correlations can evolve rapidly at the level of single genes. It might be productive to explicitly discuss your paper in these terms in the Intro.

Authors: Thanks for this suggestion! We added the following sentence to Introduction to better explain our hypothesis as well as our main result (lines 99-104):

“Using both *yfp* and *vim2*, we investigate whether trait correlations can rapidly change by mutations and selection at the level of single genes. We demonstrate that trait correlations are malleable and can undergo substantial changes on short evolutionary time scales. Although the extent of these changes varies between the proteins we study, they are predominantly affected by changes in the biophysical properties of these proteins.”

Reviewer 2. Figure 2: I can't remember what is new here and what isn't, but overall this figure is good—I particularly like showing the data from the extremes in panels C-E. But, I'm puzzled by the one red dot—could you indicate the correlation expected from mutation as a line or band in this panel, so that a reader can follow this a bit more easily?

Authors: We added a dashed line and indicated the observed trait correlation by mutations only ($R=0.63$, $p<10^{-16}$).

Reviewer 2. Figure 3 is low-res and barely readable. I think that some of these panels could be supplemental, as I feel this figure is a bit redundant; then again, a structural biologist (which I am certainly not) might really want all this detail in the main text.

Authors: Thanks for this comment. We now provide a high-resolution version of Figure 3 with our revised manuscript.

Reviewer 2. Lines 305-310: I would rephrase this to be more specific and also more positive—you're measuring correlations across genotypes, averaging over stochastic individual differences. This is very valid, but distinguishing between these levels of correlations would

help.

Authors: Thank you for this excellent suggestion! We revised the sentence to positively highlight our examination of deep mutational scanning data (lines 254-262):

“Just as in the case of fluorescent proteins, we first investigated whether mutations could change trait correlations. Assessing trait correlation in YFP and VIM2 populations requires different measurement methods: unlike light emission, antibiotic resistance cannot be measured in single cells. We thus compared trait correlations across different genotypes from deep mutational scanning experiments. This is akin to averaging the individual differences in single cells of a given genotype, and doing so for multiple genotypes. For this analysis, we analyzed data from deep mutational scans that we had previously performed to assess the fitness effect of single point mutations in VIM2 on *E. coli*'s resistance to our three antibiotics^{27,29}. “

Reviewer 2. Lines 311-330: Maybe just say once that you're measuring Spearman correlations and collect these low-information p-values in a supplemental table?

Authors: We sympathize with this suggestion, but our hands are tied by journal policy. While the repetition of p-values and statistical tests may reduce readability, we have to report the statistical test and p-values as set forth by the manuscript preparation guideline.

Reviewer 2. Lines 331-346: Is this enzyme secreted at high enough levels to constitute a public good? If so, you might not be seeing mutation-selection balance per se, but a cheater-producer balance as well. I guess this may not matter for present purposes, but I was distracted by the question.

Authors: This is an interesting though! The effect of cheater-producer balance on mutation-selection balance is indeed a good case for future investigations. Regarding VIM2, it is unlikely that this enzyme may act as a public good. We performed the experiments on agar plates where we ensured 10,000 CFUs on the plate to ensure individually visible colonies, where secretion and dissemination of VIM2 would not be possible. Previous evidence also suggests that VIM2 is poorly secreted into outer membrane vesicles in comparison to other

metallo-beta-lactamases (Nature communications, 10(1), 3617; Antimicrobial Agents and Chemotherapy, 65(10), 10-1128.).

Reviewer 2. Line 408: Missing table number?

Authors: Thanks. We added the missing number.

Reviewer 3. In this revised manuscript, Dasmeh et al. took into account many of the comments raised by the reviewers. In particular, the inclusion of new data on VIM2 beta-lactamase evolution relieved some of the original concern about the two traits of YFP (green and yellow fluorescence) being insufficiently distinct. I think the large body of data on these two proteins over several rounds of laboratory evolution, and the analysis framework to quantify the impact of mutations on two measurable phenotypes are valuable contributions to explore the overall question of how the evolvability of distant traits is related.

Authors: We appreciate the reviewer's positive assessment of our approach and the previous suggestion to study trait correlation in an enzyme which helped us generalize our observations on the evolutionary changes in trait correlation. Please find below our response to the remaining concerns.

Reviewer 3. Although I think that these contributions are sufficiently valuable to justify publication in Nature Communications, the results do not really live up to the expectations the manuscript set at the beginning talking about the evolvability of truly distinct traits in biological systems. This is because of two reasons: A) The traits the authors deal with here are too closely related and do not reflect the full spectrum of possible 'relatedness' of other traits in proteins such as K_m and k_{cat} , or oligomeric state and activity, or K_d for two co-substrates or a substrate and a cofactor, and numerous other potential examples that one can come up with. B) The take-home message of the paper – mutations that affect the foldability of a protein impact two very closely related phenotypes – is not novel; it is both one of the most intuitive solutions, and one that has been an established paradigm in the directed evolution community (including prior papers from the same group). As the reviewers have noted, the traits being discussed here are very closely related, chosen because of operational convenience, rather than their truly distinctive nature. I am not persuaded by the authors' arguments on why yellow and green fluorescence are two different traits of YFP. As I noted earlier, a fluorescent protein has a single fluorescence spectrum; just because one can measure it at two different wavelengths to get two different numbers does not make these distinct traits. Different substrates for VIM are a bit more distinct, albeit they are still very closely related to the same active site and catalytic mechanism. Now, there is nothing wrong with this approach. When dealing with a complex question like this, one needs to break it down into simpler systems that are experimentally tractable. However, the limitations of dealing with such simplified systems must be acknowledged in the discussion, instead of pretending that the question has been fully addressed for good. I encourage the authors to engage in such discussion and talk about the types of studies that would be needed to more broadly explore connections between traits that are not so closely related.

Authors: Thanks for raising these critical points. Here, we provide our response to the two concerns raised by the reviewer.

Regarding the first concern and the reviewer's point on the similarity of traits, we fully agree that it is important to explore a broad spectrum of traits with different relatedness, as suggested by the reviewer, such as broadly varying kinetic and dissociation parameters, enzymatic activities, oligomeric states, etc. One potential example of a more complex case might be the evolution of trait correlation in a bifunctional enzyme (e.g., phosphofructokinase-2, which acts both as a kinase and a phosphatase through its two different domains), or the

evolution of the correlation between the formation of dimers and tetramers in different proteins. In contrast, our systems are simpler, and their traits are more similar to each other compared to the examples provided above. Nonetheless, we respectfully submit that our protein traits fulfill the crucial criterion of being independently targeted by selection. For an analogy, consider a hypothetical scenario where a plant species exhibits both yellow and green flowers. While these color traits may be similar due to the underlying genetic factors or biochemical pathways, they serve distinct functions in attracting different pollinators. In this case, the correlation between flower colors does not indicate a lack of independence in selection pressures. Second, and in the case of VIM2 metallo-beta-lactamase, the resistance to ampicillin and cefotaxime represents two different traits which are manifested by specific subset of trait-specific residues in the active site of this enzyme.

Regarding the second concern and the take-home message of our work, we would like to elaborate on two points:

1) To our knowledge, this is the first work which shows that trait correlations can rapidly change at the level of single genes. Even without considering the molecular causes of such malleability, we think that making this point with experimental data is important and novel. This is even more so since the field of quantitative genetics has been plagued by a lack of experimental and molecular data on the evolution of trait correlations. Relatedly, our observations underscore the suitability of protein systems as valuable models for investigating and resolving longstanding questions in quantitative genetics, particularly those that have been explored in complex organisms but lacked molecular tractability.

2) We agree with the reviewer that the role of protein folding in evolution is well-established. Previous research on the relationship between protein stability and evolution has often centered around the notion that enhanced stability promotes evolvability. This argument posits that gain-of-function and potentially destabilizing mutations are more likely to become fixed in highly stable and structurally foldable proteins. However, this perspective presents a one-sided assertion, suggesting that greater protein stability invariably leads to increased evolvability. Our study offers a more comprehensive perspective. We suggest that alterations in protein stability—not solely increased stability—can make trait correlations malleable and may promote evolvability.

We are happy to follow the reviewer's suggestions to discuss these issues in the paper. First, we added a section on two key limitations of our study to the Discussion, and explained that trait relatedness might bias our conclusions (lines 435-447):

"It is important to acknowledge two key limitations of our study. Firstly, the pairs of protein traits we studied here are similar to each other. Although our selection assays independently targeted these traits, their inherent similarity may bias our conclusions, which may not apply to more dissimilar traits. Future research could explore the evolution of trait correlations in more dissimilar traits, such as enzymatic reactions catalyzed by a

bifunctional enzyme or traits related to protein oligomeric states, such as the formation of dimers and tetramers. Secondly, we relied on predicted stabilities for different variants within VIM2 populations. While stability predictors are widely used to assess protein stability^{45,54-56}, deviations from experimentally measured stabilities might affect our conclusions regarding the degree of trait correlation and its malleability. Future experiments utilizing population-level assays, such as differential scanning fluorimetry⁵⁷, may provide a better assessment of how trait correlation varies with changes in the stability or foldability of proteins.”

Second, we now elaborate on the novelty of our study regarding the connection between protein stability and evolvability (lines 355-360):

“Foldability is essential for the function of most proteins³⁷⁻⁴⁰. It can also readily change through mutations. Specifically, most random mutations in proteins are destabilizing⁴¹, but ~20% of such mutations are stabilizing and increase protein foldability⁴¹. In consequence, protein foldability is highly variable during protein evolution⁴²⁻⁴⁵. For instance, although mammalian myoglobins exhibit similar oxygen binding abilities (oxygenation constant $\sim 0.8-1.2 \mu\text{M}^{-1}$), their unfolding resistance to chemical denaturants differs dramatically, with up to a 600-fold variation^{45,46}. More generally, proteins in the proteomes of *E. coli*, *C. elegans*, *S. cerevisiae*, and human vary widely in their thermodynamic stability^{47,48}, which correlates with protein foldability³⁴. These examples suggest that changes in protein foldability occur frequently during protein evolution, potentially leading to rapid alterations in trait correlations. Our findings expand upon the established concept that highly stable and foldable proteins are more evolvable^{1,49,50}. It is not solely the increased stability or foldability of proteins that

promotes evolvability. Instead, the dynamic nature of this property can render trait correlations malleable, thereby fostering evolvability depending on whether an increased or decreased trait correlation is advantageous. “

Reviewer 3. - Line 144-146: I do not agree with this assertion. Just because a mutation shifts the fluorescence spectra of a protein a few nm in either direction, does not make emissions in these wavelengths distinct traits

Authors: We appreciate this perspective and understand the importance of clarifying our assertion. Color, indeed, plays a crucial role as a trait in various biological contexts, including fluorescence emission in proteins. While minor shifts in fluorescence spectra may not necessarily constitute entirely distinct traits, they can still represent meaningful variation in protein function or structure, and as we argued in our response to the previous point, be targeted independently by selection. We also stress that because of the same issue raised by the reviewer in the first round of review, we had included an antibiotics-conferring enzyme and resistant traits as a new study system completely different from a fluorescent protein in our analysis, and that this study system corroborated our YFP-based conclusions.

Reviewer 3. - Line 306-308: While not exactly single-cell measurement, plating the library at a dilution where each cell can give rise to a single colony does enable quantitative measurement of survivability of single cell(s) on an antibiotic challenge, thus providing an estimate of the activity of distinct mutants in these cells. In fact, this has been the basis of numerous directed evolution experiments.

Authors: Thanks for raising this point and the suggestions. We revised the paragraph in question to read more positively, as follows (lines 254-262):

“Just as in the case of fluorescent proteins, we first investigated whether mutations could change trait correlations. Assessing trait correlation in YFP and VIM2 populations requires different measurement methods: unlike light emission, antibiotic resistance cannot be measured in single cells. We thus compared trait correlations across different genotypes from deep mutational scanning experiments. This is akin to averaging the individual differences in single cells of a given genotype, and doing so for multiple genotypes. For

this analysis, we analyzed data from deep mutational scans that we had previously performed to assess the fitness effect of single point mutations in VIM2 on *E. coli*'s resistance to our three antibiotics^{27,29}. “

Reviewer 3.- Line 318: “how they”
Reviewer 3.- Line 348: Table Sx?
Line 408: Table Sx?

Authors: Thanks for catching these oversights. We fixed these points and added the table numbers.

Reviewer 3.- Overall the data from the VIM experiments used to generate the figures should be added. I do worry a little about the stability of these mutants being theoretical instead of experimental. If there is a way to confirm that, for a subset of the mutants, the theoretical and experimental values match, that would strengthen the paper.

Authors: We added all the data used in the paper either to supplementary information or to the source file provided with this submission. We agree that the experimental measurements of protein stability could be a valuable addition to the paper, but the necessary time-consuming experiments are beyond the scope of the current manuscript. However, because the concern is important, we added it as a limitation to our new discussion of limitations (lines 435-447):

“It is important to acknowledge two key limitations of our study. Firstly, the pairs of protein traits we studied here are similar to each other. Although our selection assays independently targeted these traits, their inherent similarity may bias our conclusions, which may not apply to more dissimilar traits. Future research could explore the evolution of trait correlations in more dissimilar traits, such as enzymatic reactions catalyzed by a bifunctional enzyme or traits related to protein oligomeric states, such as the formation of dimers and tetramers. Secondly, we relied on predicted stabilities for different variants within VIM2 populations. While stability predictors are widely used to assess protein stability^{45,54-56}, deviations from experimentally measured stabilities might affect our conclusions regarding the degree of trait correlation and its malleability. Future

experiments utilizing population-level assays, such as differential scanning fluorimetry⁵⁷, may provide a better assessment of how trait correlation varies with changes in the stability or foldability of proteins.”